# Tunable enantioselective electrocatalytic functionalization of unactivated alkenes

Tian Xie[1,3], Minghao Liu [1,3], Jiayin Zhang[1], Lingzi Peng [1] & Chang Guo [1,2] ✉

The tunable enantioselective functionalization of alkene feedstocks represents a highly desirable yet predominantly unresolved tactic for generating high-value scaffolds. Herein, we report a tunable enantioselective electrolytic system for dehydrogenative allylation, dehydrogenative alkenylation, and hydroalkylation reactions with identical substrates to afford structurally diverse products. This success hinges on the rational design of the stereoselective coupling of an electrogenerated nickel-bound α-carbonyl radical species that can trap unactivated alkenes and engage in various subsequent radical termination processes to enable the intermolecular functionalization of unactivated alkenes. The mild reaction conditions and sustainable electrocatalytic radical platform guarantee excellent functional group tolerance and substrate compatibility with unactivated alkenes (63 examples, up to 98% e.e.), and the process evolves $H_2$ without the need for external chemical oxidants. The utility of this enantioselective electrolytic strategy is demonstrated by its application in the stereoselective formal synthesis of (S)-SYK inhibitor, signifying substantial progress in synthetic methods.

The catalytic intermolecular functionalization of alkenes is a fundamental and essential transformation to construct basic organic frameworks in organic synthesis, which have important applications in pharmaceuticals, agrochemicals, catalysts, and materials[1]. A typical carbon−carbon bond-forming reaction, renowned as the Michael reaction[2], is the addition of carbanions or their catalytic equivalents to activated alkenes bearing electron-withdrawing groups, such as α,β-unsaturated carbonyl compounds. The catalytic functionalization of unactivated alkenes is an appealing strategy for creating structurally diverse motifs and performing late-stage derivatization of medicinally relevant molecules[3–5]. However, the low electrophilicity of unactivated alkenes makes it challenging to achieve rapid, straightforward, and stereoselective intermolecular additions with HOMO-raised nucleophiles, particularly in a tunable manner[6]. This requires alternative synthetic strategies to achieve divergent synthesis and excellent enantioselectivity[7,8]. We rationalized that radical functionalization of alkene feedstocks, such as abundant and inexpensive hexene to undergo the initial radical addition event, followed by tunable radical termination

steps to achieve multiple transformations, represents a viable strategy for enantioselective divergent alkene transformation (Fig. 1a)[9–17].

Organic electrosynthesis stands out as a sustainable synthetic avenue that enables the controlled occurrence of single electron transfer (SET) and facilitates the generation of radical intermediates from bench-stable substrates under mild conditions, eliminating the need for exogenous chemical oxidants[18–29]. However, the stereocontrolled functionalization of unactivated alkenes remains a formidable challenge. Although electrochemical oxidative allylation has been demonstrated—as exemplified by the work of the Xu group[10,12]—achieving analogous dehydrogenative alkenylation and hydroalkylation under electrolytic conditions continues to pose significant hurdles. More broadly, the development of electrochemical radical addition platforms capable of divergently accessing dehydrogenative allylation, alkenylation, and hydroalkylation from identical precursors with high chemoselectivity and stereocontrol represents a major unmet goal in synthetic chemistry. The integration of electrosynthesis and asymmetric catalysis has emerged as a highly potent and versatile

[1]Hefei National Research Center for Physical Sciences at the Microscale and Department of Chemistry, University of Science and Technology of China, Hefei, China. [2]State Key Laboratory of Coordination Chemistry, School of Chemistry and Chemical Engineering, Nanjing University, Nanjing, China. [3]These authors contributed equally: Tian Xie, Minghao Liu. ✉e-mail: guochang@ustc.edu.cn

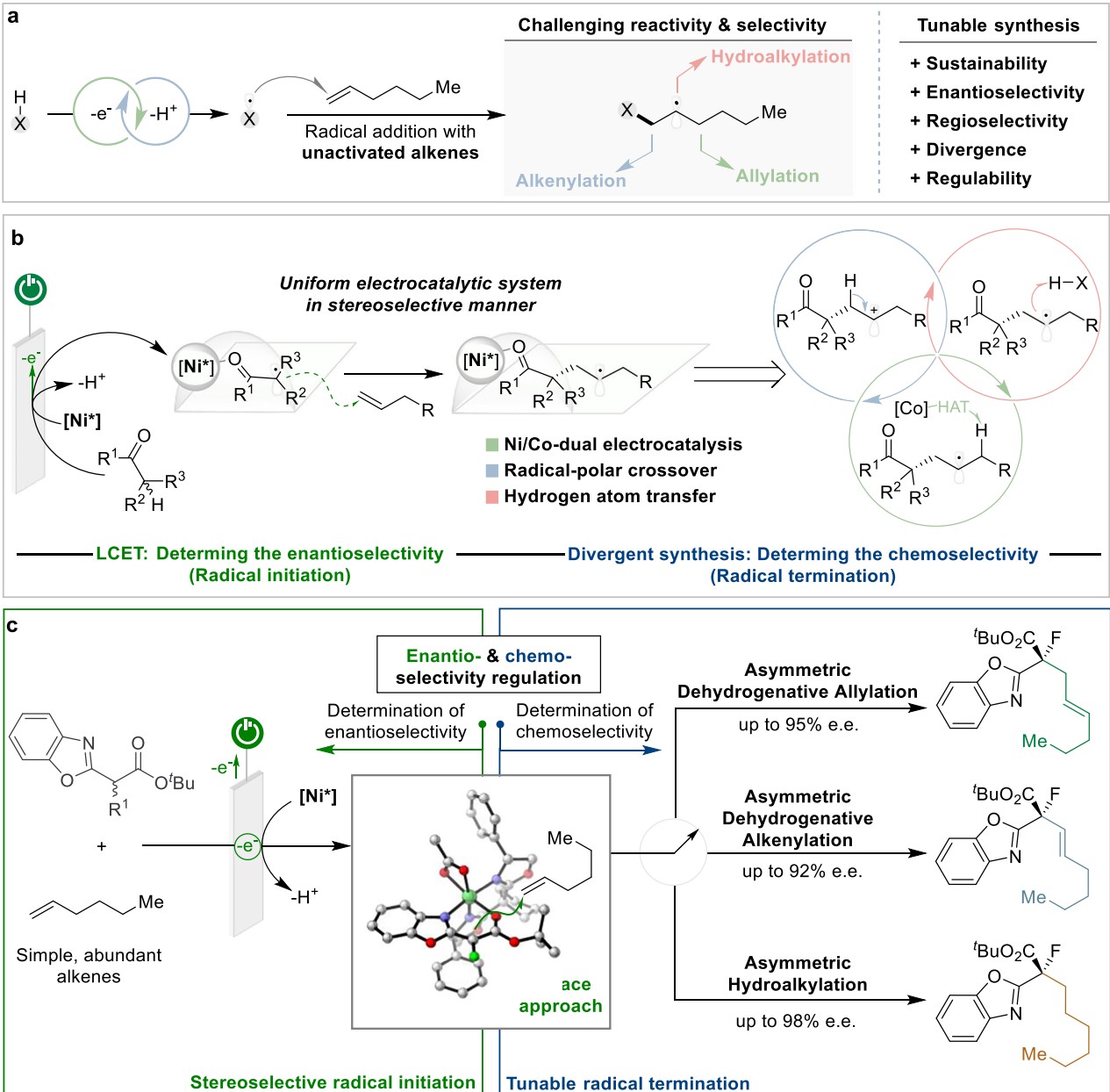

**Fig. 1 | Overview of asymmetric tunable electrocatalysis. a** Challenge and selectivity of alkene feedstock functionalization. **b** Chiral Lewis acid-coupled electron transfer (LCET) and controllable radical termination to determine the enantio- and chemoselectivity. **c** Tunable enantioselective alkene functionalizations.

synthetic approach for developing unconventional reaction pathways to generate reactive stereocontrollable radical intermediates for streamlined access to valuable chemical building blocks[30–36], offering rational and predictable means for C−C bond formation from a common precursor in synthetic chemistry[37–49].

From the outset, the endeavor to achieve tunable enantioselective functionalization of unactivated alkenes has been met with the challenge of controlling selectivity in radical addition[50–54] and termination owing to the anticipated multiple side reactions. To address this issue, we devised a catalytic cycle with a chiral Lewis acid-coupled electron transfer (LCET) process and introduced a powerful electrocatalytic platform for the enantioselective α-functionalization of carbonyl compounds (Fig. 1b)[34,35]. A vital design criterion is the nickel-bound enolate species, which are more prone to anodic oxidation, selectively generate chirals, thus effectively controlling the stereochemistry of subsequent radical addition reactions and circumventing noncatalytic racemic background reactions arising from direct oxidation of the

carbonyl substrate. Upon radical addition, we propose that the chiral electrocatalyst-mediated alkene functionalization proceeds through three competing radical termination pathways, each generating distinct products. The introduction of a secondary cobalt catalyst facilitates hydrogen atom transfer (HAT) to form dehydrogenative allylation products[14–16]. Alternatively, further oxidation of the alkyl radical enables a radical-polar crossover process to form oxidative alkenylation products[13], whereas direct hydrogen atom transfer leads to hydroalkylation products[36]. In this work, we report a general and enantioselective electrocatalytic system for the divergent functionalization of unactivated alkenes. This strategy enables dehydrogenative allylation, alkenylation, and hydroalkylation from a chiral nickel-bound radical intermediate with high stereoselectivity, broad functional group tolerance, and compatibility with diverse carbon nucleophiles (Fig. 1c). The asymmetric catalytic system provides a unified platform to systematically investigate competitive radical termination pathways under mild electrochemical conditions.

## Results

### Reaction development

To validate our hypothesis, we initiated the reactions of racemic benzoxazolyl acetate (**1a**) and hexene (**2a**) in an undivided cell equipped with a carbon felt (CF) anode and a platinum (Pt) cathode under a constant current of 2.0 mA at 25 °C (Table 1a). Encouragingly, with ferrocenemethylamine **A1** functioning as the mediator and Pybox **L1** acting as the chiral ligand, the alkenylated product **4a** could be obtained in 11% yield and 60% enantiomeric excess (e.e.). A subsequent ligand screening (**L2**-**L5**) revealed that the use of a Box ligand with a cyclopentyl group (**L5**), resulted in **4a** in 72% yield and 92% e.e. A subsequent evaluation of additives indicated that the chemoselectivity of the reaction is strongly dependent on additive selection, even when identical substrates are employed. Using the salen-Co additive **A2**, the alkenylated and allylated products were obtained with a chemoselectivity of 1:1. The similar stereoselectivities observed for both products (see Supplementary Table S2) further support that they share a common stereocontrol mode in terms of quaternary carbon center formation, whereas divergent radical termination pathways account for their structural differences. The effects of different substituted salen-Co additives on the reaction chemoselectivity were examined (**A2**-**A5**), and trifluoromethyl-substituted salen-Co **A5** selectively gave rise to allylated product **3a** (74% yield, 92% e.e.). The differential chemoselectivity among Co catalysts (**A2**–**A5**) correlates with their Co−H bond dissociation energies (BDE), where the higher BDE of **A5** enhances HAT efficiency and favors allylation product formation (see Supplementary Table S3). Notably, employing (TMS)$_3$Si-H (BDE = 79 kcal/mol) as a radical-terminating HAT reagent enabled the synthesis of hydroalkylated product **5a** in 60% yield and 93% e.e. Remarkably, all three reaction variants delivered exceptional enantioselectivity (**3a**-**5a**), underscoring the generality of this Ni-electrocatalytic approach.

Additional control experiments were carried out to elucidate the process of electricity-driven Ni/Co dual-catalyzed dehydrogenative allylation reactions (Table 1b). The absence of a chiral Lewis acid catalyst (entry 2), additive **A5** (entry 3), or electric current (entry 4) completely inhibited the generation of **3a**, thus ruling out competing racemic scenarios and highlighting the criticality of these conditions for the process. Chemical oxidation with Ag$_2$O resulted in a loss of stereocontrol, delivering racemic **3a** in 22% yield (entry 5). Additional studies revealed that product **3a** forms even without the chiral Ni catalyst (entry 6), indicating that Ag$_2$O serves a dual function as both an oxidant and a Lewis acid, thereby promoting a competing racemic pathway. The controlled electrochemical delivery of electrons ensures a clean transformation under mild redox conditions while suppressing the racemic background reactions observed with chemical oxidants (entry 1 vs 5). These results highlight the unique advantages of electrochemistry for the catalytic asymmetric transformations reported here in terms of the cleanness of conversion and enantioselectivity.

### Substrate scope

With the optimal reaction conditions established, we investigated the versatility of this electricity-enabled enantioselective dehydrogenative allylation reaction (Fig. 2). Initially, the functional group tolerance of various unactivated alkenes was tested (**3a**-**3w**). Simple alkenes bearing potentially reactive functionalities such as chloride (**3b**), bromide (**3c**), hydroxyl (**3d**), protected hydroxyl groups (**3e**-**3j**), phthalimide (**3k**), and amide (**3l**) were well accommodated in this protocol, delivering the expected products effectively. Alkene substrates bearing carbon chains of different lengths were transformed to the corresponding products in good yields, with excellent chemo- and enantioselectivities (**3m**-**3t**). The absolute configuration of **3t** was confirmed via X-ray diffraction, and the others were assigned analogously. Encouraged by the broad functional group tolerance under mild electrolytic reaction conditions, we explored the generality of the

present protocol for complex molecules. Alkenes derived from (*R*)-carvone (**3u**) and pregnenolone (**3v**) were suitable substrates. The reaction with cyclic alkene also proceeded smoothly to afford product **3w** in 70% yield. These results demonstrate that the Ni/Co dual electrocatalytic system has a wide substrate scope for unactivated alkenes with high functional group tolerance.

Further investigation with substituted racemic benzoxazolyl acetate **1** revealed that variations in the benzoxazole moiety had little effect on the reaction outcomes (**3x**-**3ab**). Variation of the ester group of racemic benzoxazolyl acetates had no noticeable effect on the reaction outcomes (**3ac**-**3af**). Racemic **1** bearing various substituents at the α-position of the carbonyl group also performed efficiently in enantioselective electrocatalytic reactions (**3ag**-**3aj**). Remarkably, the reaction also proved adaptable to 1,2-benzisoxazol-3-ylacetate, consistently delivering **3ak** in high yield and enantioselectivity (69% yield, 88% e.e.).

Upon research on electrochemically promoted asymmetric anodic allylation reactions, we demonstrate that the reaction chemoselectivity can be precisely controlled by medium engineering, where ferrocene-based **A1** selectively promotes dehydrogenative alkenylation over competing allylation pathways. Under the optimized reaction conditions, a broad range of unactivated alkenes were investigated (Fig. 3). Heteroatom-substituted alkenes proceeded smoothly under standard conditions, delivering alkenylation products with comparable yields and excellent chemoselectivities (**4a**-**4l**). Notably, the electrocatalytic dehydrogenative alkenylation approach demonstrated its practical utility for large-scale applications by effectively scaling the reaction to 6 mmol, consistently producing product **4a** in a high yield and excellent enantioselectivity. Alkenes with aryl (**4 m** and **4n**) or long-chain (**4o**) alkane substituents can effectively participate in this reaction to afford excellent enantioselectivity. X-ray diffraction analysis confirmed the *R* configuration of **4l**. Furthermore, 1,2-benzisoxazol-3-ylacetate was also compatible with the desired product in 64% yield and 92% e.e. (**4p**).

Hydroalkylation via tandem radical reactions efficiently results in the formation of quaternary carbon centers in complex molecular frameworks from simple precursors. This prompted us to investigate alternative hydrogen transfer reagents as third coupling partners to effectively trap the key alkyl radical intermediate. Having optimized the reaction conditions with (TMS)$_3$Si-H as a hydrogen transfer reagent, we investigated the generality of this asymmetric multicomponent electrochemical difunctionalization of alkenes (Fig. 4). Notably, both unactivated alkenes and diversely substituted benzoxazolyl acetates were well tolerated, consistently affording the desired products (**5a**-**5h**) in high yields with excellent enantioselectivities.

### Synthetic applications

The Ni/Co electrocatalyzed asymmetric dehydrogenative allylation reaction is also applicable to propylene gas (Fig. 5a). The alkene moiety in enantioenriched product **3a** provides a versatile synthetic handle for further transformations to afford alkane **6** and alcohol **7** (Fig. 5b). The reduction of the ester group of **3a** followed by hydrogenation of the alkene afforded corresponding alcohol **8** in 66% yield and 92% e.e. Moreover, the benzoxazole moiety in enantioenriched product **3x** was readily converted to an aldehyde group in **9** and ketone groups in **10** and **11**, enabling access to diverse chiral architectures while preserving enantiopurity.

Our enantioselective methodology was further verified via the asymmetric formal synthesis of (*S*)-SYK inhibitor (Fig. 5c)[55]. Following the reduction of enantioenriched hydroalkylation product **5d**, subsequent O-mesylation cleanly afforded corresponding dimesylate **12** while maintaining the original stereochemistry. Treatment of dimesylate **12** with benzylamine provided cyclic amine **13**, which underwent smooth N-deprotection followed by Boc protection to deliver N-Boc

## Table 1 | Optimization of tunable reaction conditions

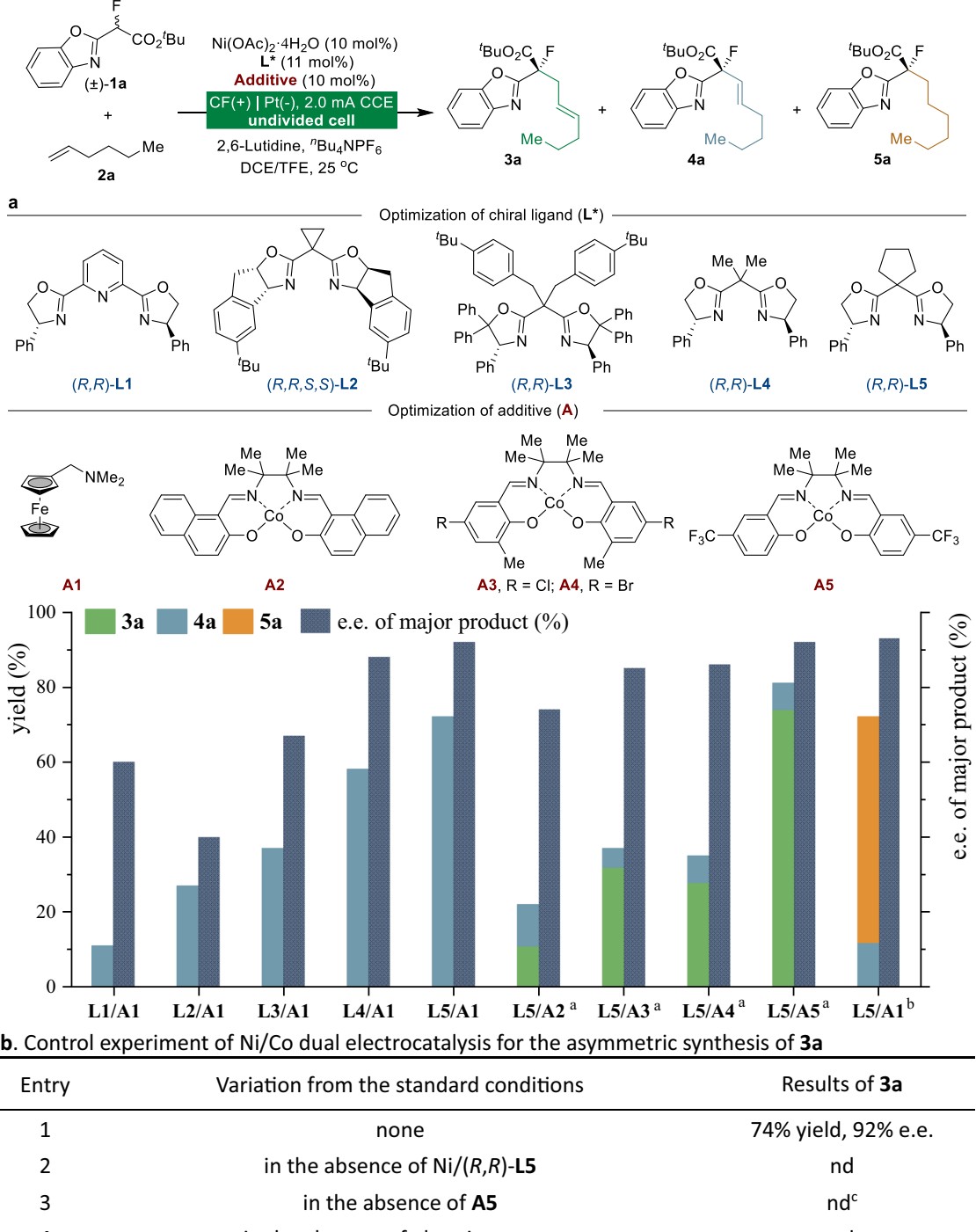

**b.** Control experiment of Ni/Co dual electrocatalysis for the asymmetric synthesis of **3a**

| Entry | Variation from the standard conditions | Results of **3a** |
|---|---|---|
| 1 | none | 74% yield, 92% e.e. |
| 2 | in the absence of Ni/(*R*,*R*)-**L5** | nd |
| 3 | in the absence of **A5** | nd[c] |
| 4 | in the absence of electric current | nd |
| 5 | Ag$_2$O (2.0 equiv.) without electricity | 22% yield, *rac* |
| 6 | Ag$_2$O (2.0 equiv.) without electricity and Ni/(*R*,*R*)-**L5** | 20% yield, *rac* |

Unless otherwise specified, all reactions were performed by using racemic benzoxazolyl acetate **1a** (0.1 mmol), alkene **2a** (0.5 mmol), Ni(OAc)$_2$·4H$_2$O (10 mol%), **L\*** (11 mol%), additive (10 mol%), 2,6-lutidine (0.1 mmol), $^n$Bu$_4$NPF$_6$ (0.1 M), DCE (0.5 mL) and TFE (2.5 mL) at 25 °C under constant-current conditions (2.0 mA, 4.5 F/mol) in an undivided cell. The enantiomeric excess (e.e.) was analyzed via high-performance liquid chromatography (HPLC). [a]Additive (20 mol%), morpholine (0.1 mmol), $^n$Bu$_4$NBF$_4$ (0.1 M) in THF/TFE, 1.5 mA, 4.5 F/mol. [b](TMS)$_3$Si-H (0.5 mmol), **2a** (2 mmol) in MeOH/TFE, 2.0 mA, 6.0 F/mol. [c]**4a** was isolated as the product with 26% yield and 92% e.e. Isolated yields after chromatography are shown. TFE, 2,2,2-trifluoroethanol; CF, carbon felt; nd, not detected.

product **14** in 92% yield with complete retention of enantiopurity. Subsequent removal of the benzoxazole group generated an aldehyde that was reduced to afford enantioenriched alcohol **15**, a versatile intermediate for (*S*)-SYK inhibitor synthesis[55].

## Mechanistic investigations

A range of experiments was conducted to explore the mechanism of asymmetric electrolytic transformation (Fig. 6). Initially, cyclic voltammetry (CV) experiments were run on the reaction components.

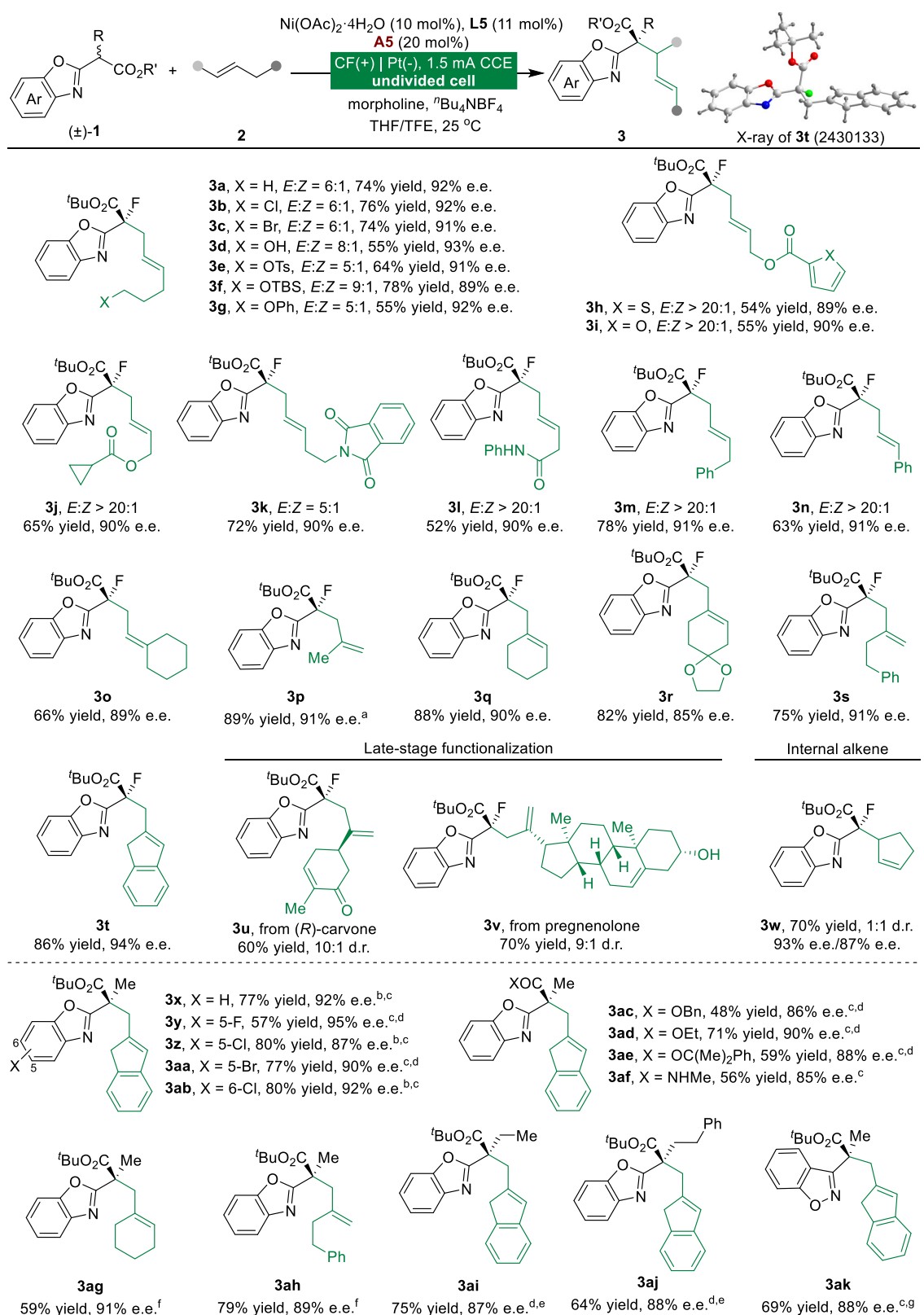

**Fig. 2 | Scope of Ni/Co dual-electrocatalyzed dehydrogenative allylation.** Reactions were performed by using **1** (0.1 mmol), **2** (0.5 mmol), Ni(OAc)$_2$·4H$_2$O (10 mol%), **L5** (11 mol%), **A5** (20 mol%), morpholine (0.1 mmol), and $^n$Bu$_4$NBF$_4$ (0.1 M) in THF/TFE under constant-current conditions (1.5 mA) in an undivided cell, 4.5 F/mol. Isolated yields after chromatography are shown. $^a$**2** (1.2 mmol). $^b$10 °C. $^c$**A2** (10 mol%), **2** (1.5 mmol) and without base. $^d$-10 °C. $^e$**A4** (10 mol%), **2** (1.5 mmol), and 2,6-Lutidine (0.1 mmol) in MeOH/TFE. $^f$**A3** (20 mol%), **L2** (11 mol%), and **2** (2 mmol), without base in DCE/TFE. $^g$**L3** (11 mol%), 2,6-Lutidine (0.1 mmol).

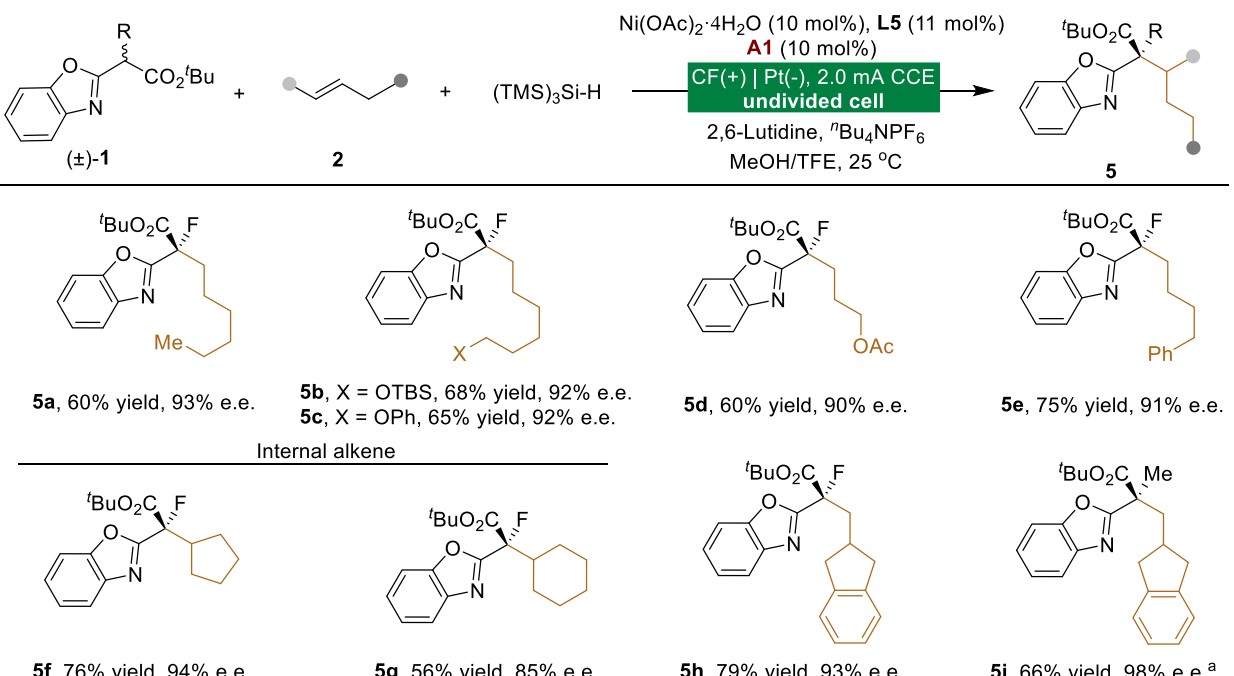

**Fig. 3 | Scope of Ni-electrocatalyzed dehydrogenative alkenylation.** Reactions were performed by using **1** (0.1 mmol), **2** (0.5 mmol), Ni(OAc)$_2$·4H$_2$O (10 mol%), **L5** (11 mol%), **A1** (10 mol%), 2,6-Lutidine (0.1 mmol), $^n$Bu$_4$NPF$_6$ (0.1 M) and DCE/TFE under constant-current conditions in an undivided cell, 4.5 F/mol. Isolated yields after chromatography are shown. [a]**L3** (11 mol%), **2** (2 mmol) at 40 °C.

**Fig. 4 | Scope of Ni-electrocatalyzed hydroalkylation.** Reactions were performed by using **1** (0.1 mmol), **2** (2 mmol), Ni(OAc)$_2$·4H$_2$O (10 mol%), **L5** (11 mol%), **A1** (10 mol%), (TMS)$_3$Si-H (0.5 mmol), 2,6-Lutidine (0.1 mmol), $^n$Bu$_4$NPF$_6$ (0.1 M) and MeOH/TFE under constant-current conditions in an undivided cell, 6.0 F/mol. Isolated yields after chromatography are shown. [a]**L2** (11 mol%) in DCE/TFE.

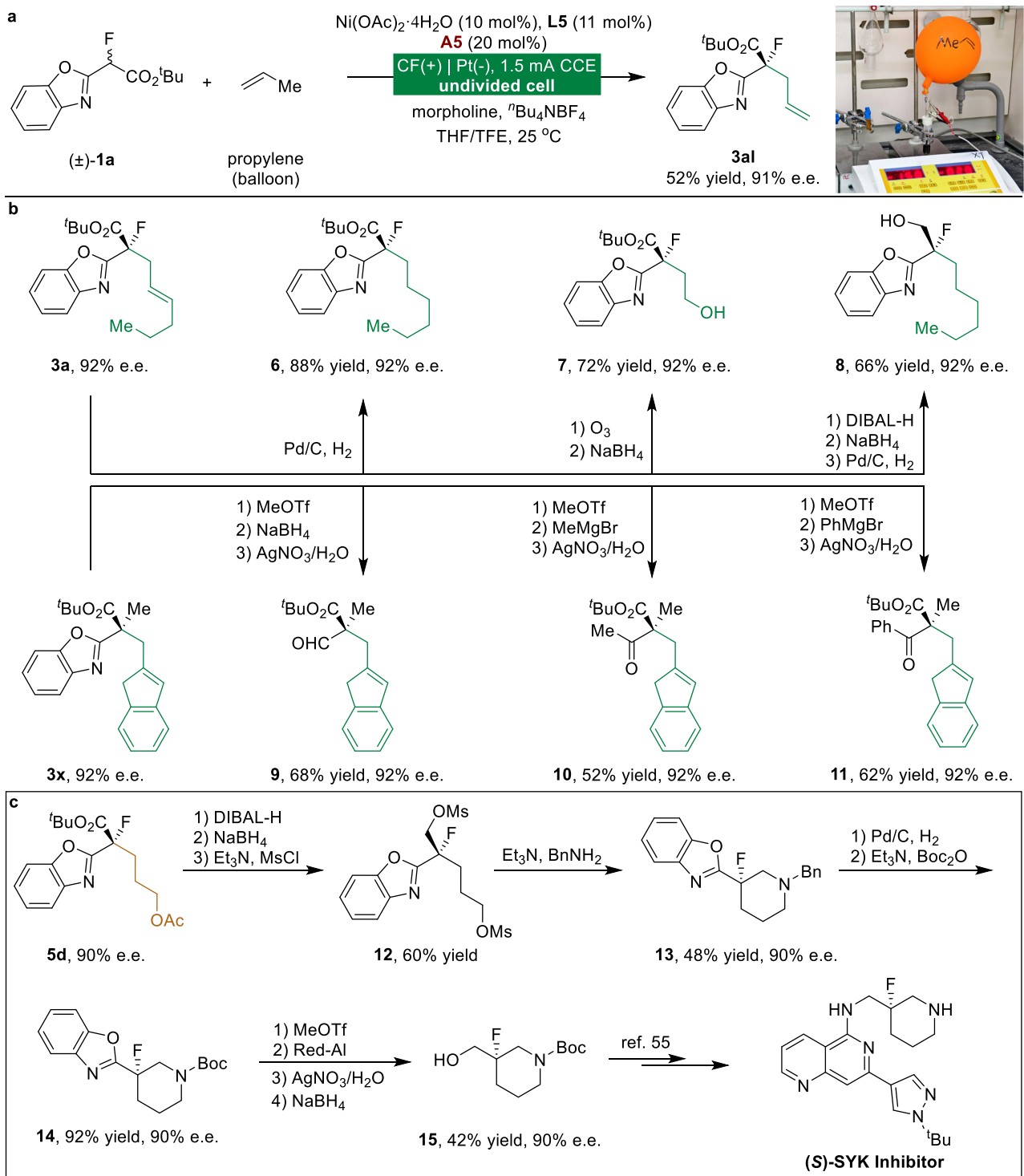

**Fig. 5 | Synthetic utility and asymmetric formal synthesis of (*S*)-SYK inhibitor. a** Asymmetric electrocatalytic direct functionalization of gaseous propylene. **b** Versatile functionalization of allylation products. **c** Asymmetric formal synthesis of (*S*)-SYK inhibitor. Isolated yields after chromatography are shown.

Upon addition of the chiral nickel catalyst, the oxidative peak of benzoxazolyl propanoate **1b** shifted negatively (Fig. 6a, left) and exhibited a concurrent increase in current with higher concentrations of **1b** (Fig. 6a, middle), supporting the facile oxidation of a nickel-bound enolate intermediate. Critically, while the substrate alone with ferrocenemethylamine **A1** showed no significant current response, the addition of the chiral nickel catalyst led to a substantial increase in the oxidative current with attenuation of the reduction peak. This clear contrast provides evidence for efficient electron transfer between **A1** and the Ni-bound enolate intermediate (Fig. 6a, right).

To unequivocally verify the hydrogen atom transfer (HAT) mechanism in the hydroalkylation reaction, we conducted a deuterium-labeling experiment by replacing (TMS)$_3$Si−H with (TMS)$_3$Si-D (Fig. 6b). The reaction exclusively afforded deuterated product **16** with 98% D-incorporation, providing direct evidence for HAT-mediated C−H bond formation and the radical nature of the key intermediate. Radical clock experiments provide further support for

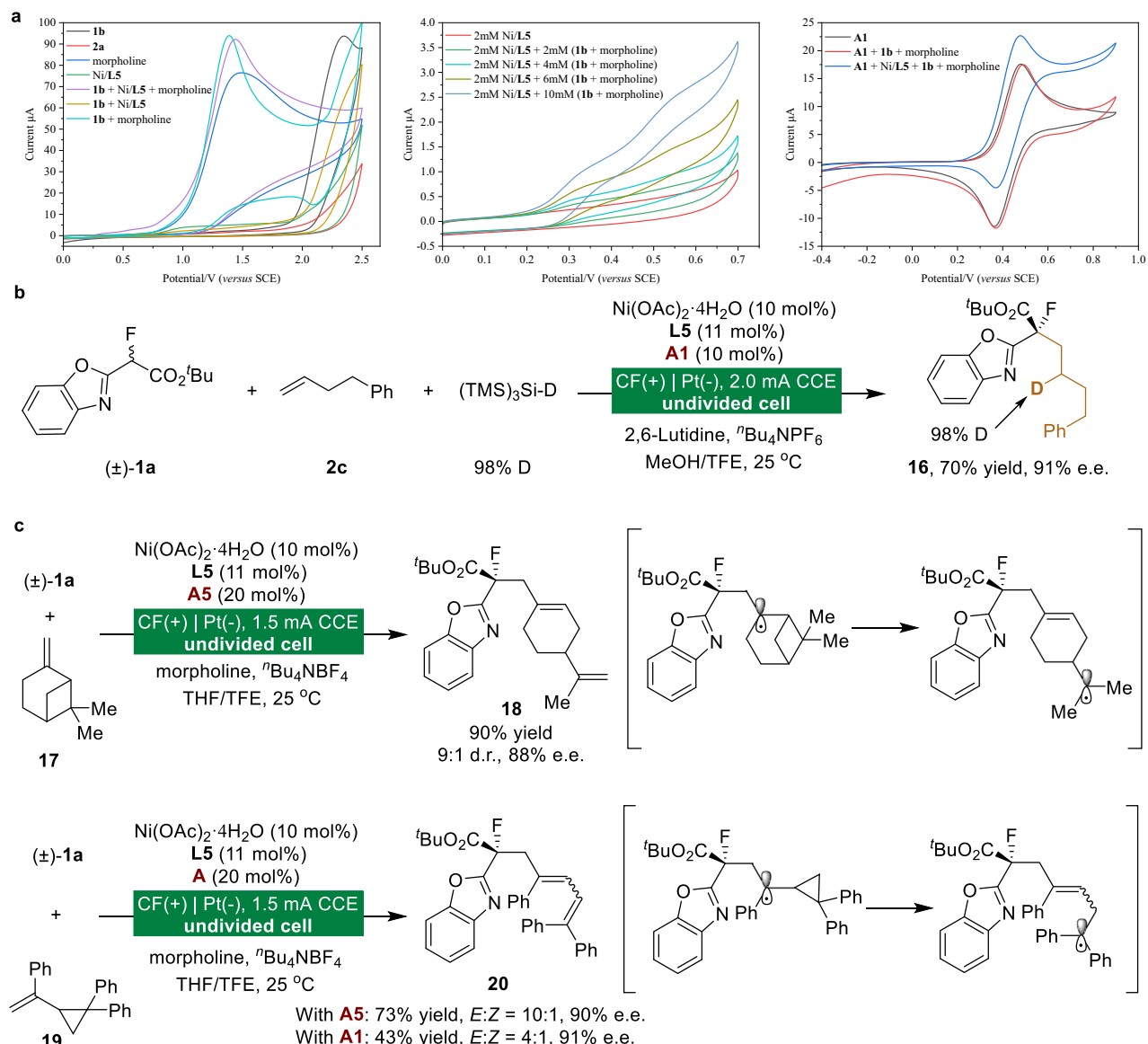

**Fig. 6 | Mechanistic investigation. a** Cyclic voltammetry of related components in the catalytic system. **b** Deuteration functionalization. **c** Radical clock experiments. Isolated yields after chromatography are shown.

the formation of the α-keto radical intermediate in the enantioselective electrochemically tunable protocol (Fig. 6c). The β-pinene **17** was treated under our electrochemical conditions, and ring-opening product **18** was obtained in 90% yield, 9:1 d.r., and 88% e.e. Furthermore, we investigated the electrochemical reaction of **1a** with cyclopropane-containing alkene **19**. Notably, both additives (**A1** and **A5**) efficiently promoted the cyclopropyl ring-opening process, delivering product **20** with excellent enantioselectivity. As supported by the data in Fig. 6, these results strongly corroborate our proposed mechanism involving nickel-bound radical intermediates in the asymmetric electrochemical functionalization of alkenes.

To unravel the origin of enantioselectivity in the Ni/Co dual electrocatalytic synthesis of allylated product **3**, we performed comprehensive DFT calculations (Fig. 7a). The computational studies systematically examined the reaction's dual selectivity that controls enantioselectivity in the radical addition step and product chemoselectivity during radical termination. Initially, the chiral nickel-bound α-carbonyl radical (**INT3**[q]) undergoes Giese addition (via **TS2-Si**[q]) to generate the alkyl radical **INT4**[q]. The transition-state structures controlling the stereochemistry are shown in Fig. 7b. **TS2-Si**[q] leads to the

experimentally preferred stereoisomer (R)−**3a**, whereas **TS2-Re**[q] produces the minor enantiomer (S)−**3a**. The difference in the free energy of activation (ΔΔG‡) was 1.8 kcal/mol, which is consistent with the experimentally observed selectivity of 92% e.e. (Table 1). The enantioselectivity primarily stems from distinct coordination modes of the benzoxazolyl ester substrate. These different coordination modes lead to opposing facial selectivities, as illustrated in Supplementary Fig. S9. Further energy decomposition analysis (Supplementary Fig. S8) indicates that the variation in coordination geometry arises mainly from steric repulsion between the phenyl ring of the substrate and the BOX ligand. We subsequently investigated the radical termination process promoted by cobalt as an HAT catalyst in the synergistic catalytic system. Radical intermediate **INT5**[d] undergoes facile β-hydrogen atom transfer (β-HAT) when activated by Co[II] species. **TS3-3a**[s(BS)] leads to the formation of dehydrogenative allylation product **3a**, whereas **TS3-4a**[s(BS)] generates dehydrogenative alkenylation product **4a**. DFT calculations reveal a 2.2 kcal/mol lower free energy barrier for HAT at the favored position (**TS3-3a**[s(BS)]) compared to the alternative site (**TS3-4a**[s(BS)]) (Fig. 7c), rationalizing the observed chemoselectivity in dehydrogenative allylation.

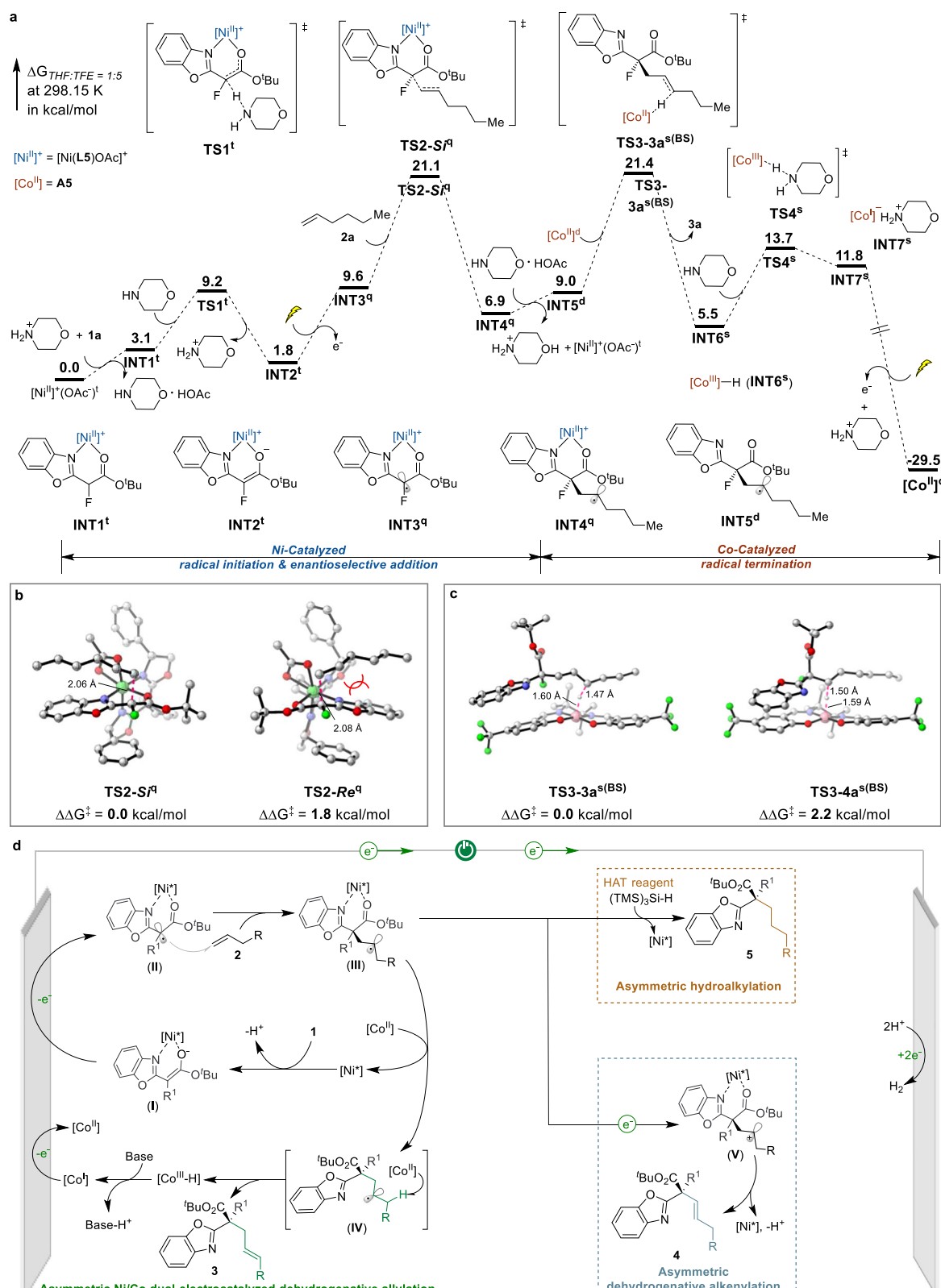

**Fig. 7 | DFT calculation and proposed asymmetric tunable electrocatalytic cycles.** All calculations described here use the wB97X-D/def2-TZVP/SMD(THF:TFE = 1:5)//B3LYP-D3(BJ)/6-31 G(d)-SDD level of theory. **a** Gibbs free energy profile of asymmetric Ni/Co dual-electrocatalyzed dehydrogenative allylation. The lightning symbol represents the anodic oxidation process, performed at +0.26 V vs. SCE corresponding to the experimentally determined onset potential of the nickel-bound enolate. Spin states are annotated as superscripts: s (singlet), d (doublet), t (triplet), q (quartet), and BS (broken symmetry). **b** Transition states of Ni-catalyzed asymmetric radical addition. **c** Transition states of Co-catalyzed HAT. For clarity, hydrogen atoms were omitted (except for H in HAT). The black numbers represent the key bond lengths in the transition states. Color scheme: C, gray; O, red; N, blue; H, white; F, bright green; Ni, light green; Co, pink. **d** Proposed catalytic cycles.

Taking into account the combined results of the mechanistic studies and DFT calculations, a plausible catalytic mechanism based on electric-triggered asymmetric catalysis is outlined in Fig. 7d. The asymmetric electrochemical catalysis is initiated by the condensation of **1** with the nickel catalyst, resulting in the formation of a nickel-bound enolate intermediate (**I**). Anodic oxidation of **I** through an anodic SET pathway was proposed to form a nickel-coordinated radical intermediate (**II**). This radical species undergoes stereoselective interception by unactivated alkenes to establish a stereogenic center, forming alkyl radical **III**, a key step that governs asymmetric induction in these tunable catalytic cycles. Critically, the divergence between the dehydrogenative allylation, dehydrogenative alkenylation, and hydroalkylation pathways is governed by distinct radical termination processes. Under Ni/Co dual electrocatalytic conditions, in-situ generated alkyl radical **III** undergoes a sterically controlled HAT process via the cobalt catalyst, preferentially forming dehydrogenative allylation products **3** at the less hindered site. Alternatively, the introduction of (TMS)₃Si-H as the group-transfer reagent selectively traps radical **III** to deliver hydroalkylation products **5**. Conversely, the use of additive **A1** in the absence of additional termination reagents promotes single-electron oxidation of alkyl radical intermediate **III** to form carbocationic species **V**. This intermediate is stabilized by 2,6-lutidine and undergoes base-promoted β-hydride elimination preferentially from the more acidic proximal carbon, ultimately furnishing enantioenriched dehydrogenative alkenylation products **4**.

We have developed a versatile nickel-electrocatalytic system that enables tunable enantioselective transformations of identical substrates to access three distinct product classes—dehydrogenative allylation, dehydrogenative alkenylation, and hydroalkylation—all with excellent enantioselectivity (up to 98% e.e.). The key to this success is the stereocontrolled coupling of electrogenerated nickel-bound α-carbonyl radicals with unactivated alkenes, where selective radical termination pathways dictate chemoselectivity. This sustainable electrocatalytic platform operates under mild conditions without external oxidants, demonstrating exceptional functional group tolerance and a broad substrate scope (63 examples). The synthetic utility of this approach is highlighted through a streamlined formal synthesis of the (*S*)-SYK inhibitor, indicating a significant advance in electrochemical asymmetric synthesis.

## Methods

### General procedure for the enantioselective Ni/Co dual-electrocatalytic functionalization of unactivated alkenes

A 10 mL flask equipped with a magnetic stir bar was charged with **1** (0.1 mmol), **2** (0.5 mmol), Ni(OAc)₂·4H₂O (0.01 mmol), **L5** (0.011 mmol), **A5** (0.02 mmol), morpholine (0.1 mmol), $^{n}$Bu₄NBF₄ (0.3 mmol), THF (0.5 mL) and TFE (2.5 mL) under an argon atmosphere. The flask was equipped with a carbon felt (1.5 cm × 1.5 cm × 3 mm) as the anode and a platinum plate (1.0 cm × 1.0 cm × 0.2 mm) as the cathode. Constant current (1.5 mA) electrolysis was carried out at 25 °C until complete consumption of the starting material (monitored by TLC). The solvent was removed under reduced pressure. The residue was purified by silica gel chromatography to afford the desired product **3**.

## Data availability

Crystallographic data for the structures reported in this article have been deposited at the Cambridge Crystallographic Data Centre, under deposition numbers CCDC 2430133 (**3t**) and CCDC 2430132 (**4l**). Copies of the data can be obtained free of charge via https://www.ccdc.cam.ac.uk/structures/. All other data supporting the findings of this study, including experimental procedures and compound characterization, NMR, and HPLC are available within the Article and its Supplementary Information or from the corresponding authors. The authors declare that all other data supporting the findings of this study are available within this Article and its Supplementary Information.

Source Data are provided with this paper. NMR data in a Mnova file format and HPLC traces are available at Zenodo at https://zenodo.org/records/17364427, under the Creative Commons Attribution 4.0 International license. Source data are provided with this paper.

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

## Acknowledgements

The authors thank the supercomputing center of USTC for providing computational resources and the Instruments Center for Physical Science of USTC. The authors acknowledge financial support from the National Key R&D Program of China (2023YFA1506700, C.G.), the National Natural Science Foundation of China (grant no. 21971227, 22222113, C.G.), CAS Project for Young Scientists in Basic Research (YSBR-054, C.G.), the Fundamental Research Funds for the Central Universities (WK9990000090, WK9990000111, C.G.), and the Fundamental Research Funds for the Central Universities (WK9990000133, C.G.). The project was supported by the Open Research Fund of the State Key Laboratory of Coordination Chemistry, School of Chemistry and Chemical Engineering, Nanjing University, China Postdoctoral Science Foundation (grant no.2023M743373, 2024T170881, L.P.), and the Postdoctoral Fellowship Program of CPSF (GZB20230708, L.P.).

## Author contributions

C.G. conceived the project. T.X. performed the experiments and analyzed the data. M.L., J.Z., and L.P. synthesized the substrates and ligands. All authors discussed the results and prepared the manuscript.

## Competing interests

The authors declare no competing interests.
