## [Transparent Peer Review file · Nature Communications]

Tunable Enantioselective Electrocatalytic Functionalization of Unactivated Alkenes

Corresponding Author: Dr Chang Guo

Version 0:

Reviewer comments:

Reviewer #1

(Remarks to the Author)

Comment: Guo et al. report a divergent nickel-catalyzed enantioconvergent functionalization of unactivated alkenes, enabling switchable access to dehydrogenative allylation, dehydrogenative alkenylation, and hydroalkylation. The success of this strategy hinges on the rational design of an electro-oxidatively generated nickel-bound α -carbonyl radical species, which adds to unactivated alkenes and undergoes diverse radical termination pathways. The developed protocols exhibit broad substrate scope, excellent chemoselectivity, regioselectivity, and enantioselectivity, as well as scalability. The method is successfully applied to the stereoselective formal synthesis of an (S)-SYK inhibitor. Mechanistic and computational studies provide support for the proposed reaction pathway. This work represents a significant advance in asymmetric electrocatalysis and is a substantial contribution worthy of publication.

1. The authors propose an indirect electro-oxidative generation of the nickel-bound α -carbonyl radical species mediated by a CoIII complex in allylation. What evidence rules out a direct anodic oxidation pathway since the alkenylation and hydroalkylation process do not need the CoIII complex?
2. Besides quaternary carbon stereocenters, could these protocols construct related tertiary stereocenters? If not, what is the reason?
3. For hydroalkylation process, the involvement of NiH from Ni precatalyst and (TMS)₃SiH is well documented. Did the authors have any experimental evidence to rule out this possibility?

Reviewer #2

(Remarks to the Author)

Guo and co-workers report an enantioselective functionalization of unactivated alkene with benzoxazolyl acetate via electrochemically generated carbon-centered radicals. These radicals are stabilized by a chiral Ni catalyst, and subsequent enantioselective radical addition to alkenes—followed by hydrogen atom transfer (HAT), further oxidation, or reduction with hydrosilane—yields products of dehydrogenative allylation, alkenylation, and hydroalkylation, respectively. This methodology enables the formation of highly enantioenriched products, including those bearing all-carbon quaternary stereocenters. The authors also provide detailed mechanistic insight into the chemo- and enantioselectivity based on cyclic voltammetry and DFT calculations.

However, similar transformation by the same group have already been reported (Refs. 34 & 35), and radical addition to alkene moieties are well-established in prior literature (Refs. 10–12). Therefore, the key novelty of this work appears to lie in the switchable radical termination strategy. This reviewer recommends that the authors clarify the individual novelty and significance of each transformation pathway—namely, dehydrogenative allylation, alkenylation, and hydroalkylation. A more explicit discussion will help readers better appreciate the scope and innovation of the work. This report may be perceived as merely a combination of a known radical reaction in the first half and a known termination reaction in the latter half. The reviewer believes it is essential to provide a clear and thorough discussion on what advantages this methodology offers. Furthermore, the origin of the selectivity, particularly between the allylation and alkenylation pathways, requires further clarification. According to the authors, the use of a Co catalysts leads to allylation, possibly due to steric hindrance influencing HAT. In contrast, oxidation followed by deprotonation results in alkylation. However, this reviewer finds the explanation for the selective deprotonation step insufficient. What governs the divergence between these two pathways? A more detailed mechanistic rationale, possibly supported by additional data or calculations, would greatly enhance the manuscript.

Additional points:

> In all electrochemical reactions, the total passed (in F/mol) should be explicitly stated. Since external oxidants are rarely replaced by anodic electron transfer, the charge corresponds directly to the equivalent of oxidant used. This value can be calculated from the current and reaction time and should be reported for each transformation.

> Page 1, line 25: "Main text" appeared to be a typographical error and should be corrected to "Introduction"

> Page 8 line 199, The phrase "alcohol 7 with all-carbon quaternary stereocenters" is incorrect, as compound 7 does not possess an all-carbon quaternary stereocenter.

> In the CV analysis, the authors note a significant decrease in oxidation potential upon addition of the Ni catalyst. However, the oxidation of morpholine also occurs at a similarly low potential (Figure 6a, top-left). To clearly distinguish the contribution of the Ni catalyst from that of morpholine, the cyclic voltammogram of a mixture of 1b and Ni catalyst should be measured and presented.

Reviewer #3

(Remarks to the Author)

Reviewer #4

(Remarks to the Author)

The manuscript entitled "Enantioselective Switchable Selectivity in Electrocatalyzed Functionalization of Unactivated Alkenes" by Guo et al. describes the development of the asymmetric electrochemical α -functionalization of benzoxazolyl acetate derivatives with alkenes, via the use of a chiral Ni(II) complex. This method permits the transient formation of a chiral carbon-centered radical intermediate, able to diastereoselectively couple with alkenes to generate alkyl radicals which, depending on the reaction conditions, lead to the formation of alkenyl-, allyl-, or alkyl-functionalized products.

The reaction outcome depends on the nature of the catalyst or the reagent introduced in the reaction mixture. Employing the ferrocene derivative A1 as redox mediator alone preferentially leads to the formation of the alkenylated products, where the cobalt complex A5 permits the selective preparation of the corresponding allylated derivatives. Alternatively, when A1 is employed in the presence of $(\text{Me}_3\text{Si})_3\text{Si-H}$ as terminal HAT reagent, the corresponding alkylated products are formed. The manuscript is very well written and the supporting information is of excellent quality.

The development of enantioselective electrocatalytic transformations is of high interest in organic chemistry and is still a difficult goal to achieve. In this area, the use of chiral Lewis acids or chiral organocatalysts has recently emerged as the most promising catalytic strategies. In line with previous work from Guo et al. concerning the enantioselective electrochemical amination, α - and γ -nitroalkylation, difunctionalization of terminal alkynes, and functionalization of alkenes, the strategy which is employed is based on the single electron oxidation of catalytically generated chiral nickel(II) enolate species, and the subsequent capture of the electrophilic radical which is formed by an alkene. Somehow, it can be regarded as a natural extension of previous work with one another type of radical quencher.

In the introduction of the manuscript, the authors emphasize the term "switchable," seemingly as a way to highlight the versatility of their methodology in delivering different products. However, the use of this term may be somewhat misleading. In the context of catalysis, "switchable catalysis" typically refers to processes in which the reaction outcome can be altered by applying an external stimulus (e.g., light, heat, pH, or changes in electronic state). In the present case, the different transformations— α -alkenylation, α -allylation, and α -alkylation—require distinct reaction conditions rather than a single system being modulated by an external factor. Therefore, the use of the term "switchable" may not be entirely appropriate in this context.

The optimizations of the reaction's conditions are rather succinct, it would be more informative to discuss the exact role of the DCE/TFE solvent mixture, why sometimes 2,6-lutidine is used and, sometimes, morpholine is preferred. How can we rationalize the fact that A5 is better than A2-A4, what is the reactivity difference of A5 vs A1 (see below)?

With regard to the mechanistic studies performed, several important questions remain unanswered.

First, why were all cyclic voltammetry (CV) experiments conducted in the presence of morpholine rather than lutidine? Lutidine is less prone to oxidation than morpholine, and this difference could significantly impact the CV results. It is worth noting that while lutidine is employed in both the alkenylation and alkylation reactions, morpholine is used exclusively in the Co-catalyzed allylation process.

Additionally, the experimental details for the CV measurements are insufficient. Key parameters such as the concentrations of analytes (as a result, the stoichiometry of the various analytes in the cell cannot be determined, which is critical considering the significant current measured), the solvent system, and the scan rate are not reported. Furthermore, the rationale for selecting compound 1b as the model substrate instead of 1a is not explained.

More critically, the conclusions drawn by the authors are questionable. The oxidation wave attributed to the nickel(II) enolate appears very close to that of morpholine (Figure 6a, top left). Given that the reaction conditions involve a tenfold excess of morpholine relative to the chiral nickel catalyst (which is likely not the case for the CV measurements), competitive oxidation of morpholine cannot be ruled out. The earlier onset of the oxidation wave upon increasing the concentration of nickel catalyst (Figure 6a, top right) is not sufficient evidence to confirm the oxidation of the nickel enolate. The measured currents are very weak, and once again, it appears that only one equivalent of morpholine was used.

Even more problematic is the interpretation of the role of the Co(II)-salen catalyst. The authors suggest that the

disappearance of the reverse peak indicates oxidation of the nickel enolate by the Co(III) complex via a single-electron transfer (SET) process, thereby generating a carbon-centered radical. However, the CV data do not support this conclusion. Specifically, increasing the concentration of the nickel catalyst does not result in a catalytic wave (Figure 6a, bottom right). If Co(III)⁺ generated electrochemically were indeed oxidizing the enolate via SET, a catalytic current—similar to that observed with compound A1—should be present. Since it is not, this suggests that a two-electron oxidation is taking place, or that a chemical transformation other than a simple electron transfer is occurring between the nickel enolate and Co(III)⁺. This finding contradicts the authors' proposed mechanism and raises serious doubts about the validity of the allylation pathway as described. Consequently, the DFT calculations based on a SET mechanism are also questionable. In addition, the proposed regeneration of the Co(II) catalyst is presented inconsistently. In Figure 7a, it is suggested via DFT calculations to occur via comproportionation between Co(I) and Co(III), whereas Figure 7d instead depicts anodic oxidation as the operative step. This discrepancy should be clarified.

In conclusion, the manuscript by Guo et al. presents interesting findings but lacks sufficient novelty to be considered highly innovative. The work appears to be a natural extension of previous studies, employing alkenes to trap a transient chiral, carbon-centered radical intermediate. While the authors claim that the reaction conditions allow access to three types of products—alkenylated, allylated, or alkylated—the description of the process as "switchable" is misleading. Rather than a single switchable process, the study describes three distinct reactions proceeding via closely related radical pathways, with alkenes serving as radical traps (reminiscent of their previous work: *Sci. Adv.* 2022, 8, eadd7134).

The fate of the transient alkyl radical follows well-established transformations, including oxidation/elimination, oxidative hydrogen atom transfer (HAT) with a cobaloxime-type catalyst, and reductive HAT using (Me₃Si)₃Si-H. However, the proposed mechanism is questionable, particularly regarding the role and mode of action of the Co(II)-salen catalyst. Cyclic voltammetry (CV) experiments suggest that the proposed single-electron transfer (SET) oxidation of the nickel enolate by electro-generated Co(III)⁺ is unlikely.

For these reasons, I do not recommend publication of this manuscript in *Nature Communications*. Instead, I suggest the authors revise the mechanistic rationale and consider submitting the work to a more specialized journal after addressing these concerns.

Reviewer #5

(Remarks to the Author)

I co-reviewed this manuscript with one of the reviewers who provided the listed reports. This is part of the *Nature Communications* initiative to facilitate training in peer review and to provide appropriate recognition for Early Career Researchers who co-review manuscripts.

Reviewer #6

(Remarks to the Author)

In this manuscript, Guo et al. reported an enantioselective switchable electrolytic system for the functionalization of unactivated alkenes, which enables dehydrogenative allylation, dehydrogenative alkenylation, and hydroalkylation reactions with identical substrates to produce structurally diverse products. The success of this system relies on the stereocontrolled coupling of an electrogenerated Ni-bound α -carbonyl radical with alkenes, followed by divergent radical termination pathways. The method operates under mild, oxidant-free conditions, exhibits broad substrate scope (63 examples), and delivers high enantioselectivity (up to 98% ee). Applications include late-stage functionalization of complex molecules and a formal synthesis of (S)-SYK inhibitor, highlighting its synthetic utility. Mechanistic studies and DFT calculations support the proposed catalytic cycle. The following issues should be considered:

1. In the optimization of switchable reaction conditions, the authors mentioned the chemoselectivity of the reaction is strongly dependent on additive selection. It could be seen that the additive A1 with any ligand could form the alkenylated product 4a, while the presence of salen-Co additives could achieve the allylated product 3a. This is really interesting. DFT calculations partly explain the role of the Co additives, the additive A1 should also be calculated to clarify the chemoselectivity of this reaction. It would be better if the different effects of salen-Co additives for the diversity in ratio of different chemoselective products could be explained.
2. The ratios of E/Z product was mentioned in this manuscript, it would be great if the authors could also consider the possibility of E/Z product formation in the reaction.
3. Did the authors consider the spin states of Co and Ni in the DFT calculation? The authors only mentioned "For transition states TS-3a and TS-4a, the 'broken symmetry' formalism was used" in Supporting Information, have the authors tried and compared different spin states for the key stationary points? It is weird that different spin states were used for stationary points along reaction pathways in the calculation without additional explanation. This should be described in the manuscript including Figure 7(a).
4. The origin of stereoselectivity could be explained by NCI or EDA analysis. It is hard to understand TS2-Si is more favorable form the structures. Considering that TS2 is the key transition state structure that determines stereoselectivity, the discussion on its conformation in Figure S8 should be briefly mentioned in the manuscript.
5. The structure of 2a should be given in Figure 7.

Version 1:

Reviewer comments:

Reviewer #1

(Remarks to the Author)

All my concerns have been addressed and I does not have further concerns.

Reviewer #2

(Remarks to the Author)

I have reviewed the revised version of the manuscript. Overall, the authors have addressed the points raised in the previous review, and the manuscript is much improved.

Although the cyclic voltammetry experiments may require further careful consideration, as also noted by Reviewer 4, this reviewer looks forward to further studies by the authors on the radical generation mechanism.

Therefore, this reviewer is happy to recommend the manuscript for publication as is.

Reviewer #3

(Remarks to the Author)

Reviewer #4

(Remarks to the Author)

After proofreading this manuscript for the second time, I acknowledge that this new version has been significantly improved and has addressed most of my previous comments. Notably, the revised mechanistic proposal, excluding any role of Co(III) as a redox mediator in the oxidation of the Ni-enolate through a SET process, is more reasonable. The analyses and DFT calculations now better support the observed selectivity.

However, in my opinion, several aspects of the electrochemical studies remain unclear. To ensure relevant electrochemical measurements, the supporting electrolyte should be present in large excess compared with the substrates. In the current work, the concentration of the supporting electrolyte (50 mM) is only about twice that of 1b and the amine (33 mM), which is insufficient. Increasing the supporting electrolyte concentration could improve, even change, the CV profiles.

The concentrations used for each component are too high. Considering the relatively large electrode (3 mm diameter), it would be more appropriate to present current values in microamperes (μA) rather than amperes (A). In the present case, the reported currents appear low but should be around 10 μA , which is much more realistic than the 0.3 mA values shown. Regarding the determination of onset potentials in the Supporting Information, this point deserves closer examination, as it is used to validate the authors' mechanistic discussion. The onset potential for morpholine appears to be below 1.16 V. The onset should be determined at the point where the current deviates from zero — with such high currents, demonstrating oxidation should not be problematic. The cyclic voltammograms (CVs) presented are not entirely convincing due to the excessively high currents observed for the 1b + morpholine mixture. A more convincing approach would be to record CVs with increasing amounts of this mixture added to a 1–2 mM Ni/L5 solution, yielding current intensities in the μA range. The authors state that “the initial oxidative potential of benzoxazolyl propanoate 1b decreased significantly from +1.92 V to +0.32 V (versus SCE) upon addition of the chiral nickel catalyst.” Such comparisons are inappropriate; instead, the oxidation potential of the enolate (1b + morpholine) should be compared to that of the Ni-enolate. In this respect, the black arrow in Figure 6a is confusing. Moreover, the CVs obtained with increasing catalyst concentrations are not relevant and, in my opinion, should be removed from the main text. Under the reaction conditions, only about 10 mol% of the nickel enolate is present alongside ten equivalents of the enolate; thus, the middle panel of Figure 6a does not provide meaningful information and should likely be omitted.

The combination of only two reaction partners shown in the rebuttal is very informative. However, the CV of 1b with Ni alone was not presented. It would be helpful to include this binary combination, in the Supporting Information.

The authors claim that both 2,6-lutidine and morpholine can promote the alkenylation and alkylation reactions, yet CV data were only provided for morpholine. They justify this by stating that morpholine affords a more distinct catalytic current and clearer interpretation of the electrochemical response. However, no catalytic current is observed with either morpholine or 2,6-lutidine in the comparison CVs shown in the rebuttal. This explanation is therefore unconvincing and should be reconsidered. The only compelling observation is that displayed in Figure 6a (right), which shows that A1 can act as a redox mediator. Nevertheless, this role is not integrated into the mechanism proposed in Figure 7. It would be useful to perform CVs under conditions closer to the catalytic system — namely with a 1:1 mixture of A1 and Ni/L5, along with ten equivalents of 1b and 2,6-lutidine — rather than the conditions used in Figure S8, which are not representative.

In conclusion, since the stereocontrolled electrochemical functionalization of unactivated alkenes remains a largely unexplored and challenging area, I believe this work is of significant interest to the scientific community. However, the electrochemical data supporting the proposed mechanism require further improvement. For these reasons, I recommend publication in Nature Communications once the authors have thoroughly addressed all the issues mentioned above.

Reviewer #5

(Remarks to the Author)

Reviewer #6

(Remarks to the Author)

The authors have revised the manuscript according to my comments, it could be acceptable now.

Version 2:

Reviewer comments:

Reviewer #4

(Remarks to the Author)

I have examined the revised version of the manuscript, which now includes additional cyclic voltammetry (CV) experiments. The authors have satisfactorily addressed most of the concerns raised in the previous review, and the overall quality of the manuscript has improved considerably. Nonetheless, in their rebuttal letter, the authors continue to use the term "catalytic current," which is not appropriate in the present context. Demonstration of a genuine catalytic process would require a much more pronounced effect, rather than the modest increase in oxidation peak intensity accompanied by an almost negligible decrease in the reduction peak. In any case, since the revised manuscript no longer refers to a catalytic current, this issue can be considered resolved.

Reviewer #5

(Remarks to the Author)

Responses to Comments

for

Tunable Enantioselective Electrocatalytic Functionalization of Unactivated Alkenes

Tian Xie, Minghao Liu, Jiayin Zhang, Lingzi Peng, Chang Guo[✉]

Manuscript ID: NCOMMS-25-46490-T

Reply to comments by Reviewer 1

We appreciate **Reviewer 1** for favorable comments!

- (1) Guo et al. report a divergent nickel-catalyzed enantioconvergent functionalization of unactivated alkenes, enabling switchable access to dehydrogenative allylation, dehydrogenative alkenylation, and hydroalkylation. The success of this strategy hinges on the rational design of an electro-oxidatively generated nickel-bound α -carbonyl radical species, which adds to unactivated alkenes and undergoes diverse radical termination pathways. The developed protocols exhibit broad substrate scope, excellent chemoselectivity, regioselectivity, and enantioselectivity, as well as scalability. The method is successfully applied to the stereoselective formal synthesis of an (S)-SYK inhibitor. Mechanistic and computational studies provide support for the proposed reaction pathway. This work represents a significant advance in asymmetric electrocatalysis and is a substantial contribution worthy of publication.

Answer: We appreciate Reviewer 1 for the favorable comments and helpful suggestions! These comments are greatly valuable and helpful for revising and improving our manuscript. We have made all the necessary amendments as suggested in our revised manuscript and Supplementary Information.

- (2) The authors propose an indirect electro-oxidative generation of the nickel-bound α -carbonyl radical species mediated by a CoIII complex in allylation. What evidence rules out a direct anodic oxidation pathway since the alkenylation and hydroalkylation process do not need the CoIII complex?

Answer: Upon careful examination, we confirm that the allylation pathway does not necessarily require mediation by the CoIII complex. Control experiments show that in the absence of the cobalt mediator, the alkenylation product **4a** can be obtained in 26% yield via direct anodic oxidation (Table 1b, entry 3), which clearly demonstrates that the nickel-bound α -carbonyl radical species can be generated at the anode.

We have revised our mechanistic proposal and computational analysis. The updated mechanism now features direct anodic oxidation of the nickel enolate as the reaction pathway for radical generation. The role of the cobalt catalyst has been redefined to specifically facilitate the hydrogen atom transfer step in the allylation pathway, rather than participating in the initial oxidation event. Accordingly, we have revised the theoretical calculations (Fig. 7a) and proposed catalytic cycles (Fig. 7d) in the revised manuscript.

(3) Besides quaternary carbon stereocenters, could these protocols construct relate tertiary stereocenters? If not, what is the reason?

Answer: We evaluated substrate **S1** under our standard conditions for allylation, alkenylation, and hydroalkylation reactions. Unfortunately, none of the desired products were detected.

We attribute the lack of reactivity to two primary factors. First, the radical intermediate generated from substrate **S1** is less sterically hindered and consequently more reactive. This high reactivity leads to unproductive side pathways, such as dimerization and over-oxidation, which was corroborated by reaction monitoring and product analysis revealing multiple byproducts. Second, and more critically, the stereocontrol in our method hinges on the dynamic kinetic resolution (DKR) of a trisubstituted benzoate species. The tertiary carbon-centered radical generated from **S1** would not undergo facile racemization required for an efficient DKR process. Consequently, even if the radical could be generated, achieving enantiocontrol within our current catalytic framework would be unfeasible. Therefore, our protocol is not suitable for constructing enantioselective tertiary stereocenters. We have included these results in the

revised Supplementary Information (Page S13, Table S5).

- (4) For hydroalkylation process, the involvement of NiH from Ni precatalyst and (TMS)₃SiH is well documented. Did the authors have any experimental evidence to rule out this possibility?

Answer: Thank you for raising this point regarding the potential involvement of a Ni-H species. We agree that a Ni-H pathway is a well-established mechanism in nickel-catalyzed hydroalkylation.

To evaluate the potential involvement of a Ni-H species in our system, we conducted a control experiment under the standard hydroalkylation conditions, replacing (TMS)₃SiH with (MeO)₃SiH—a hydrogen atom donor widely used in and characteristic of Ni-H-mediated processes (*Angew. Chem. Int. Ed.* **2022**, *61*, e202208018; *J. Am. Chem. Soc.* **2023**, *145*, 10411; *Nat. Synth.* **2024**, *3*, 1360; *J. Am. Chem. Soc.* **2025**, *147*, 18944; *Sci. Adv.* **2025**, *11*, eadv6571). DFT calculations confirm that the Si-H bond dissociation energy of (MeO)₃SiH is significantly higher than that of (TMS)₃SiH. Furthermore, the observed inactivity of (MeO)₃SiH under our reaction conditions corroborates our earlier finding that a low BDE is essential for the reaction to proceed (*J. Am. Chem. Soc.* **2025**, *147*, 8917). These computational and experimental results collectively support a hydrogen atom transfer (HAT) mechanism, rather than one involving a Ni-H intermediate. We have included these results in the revised Supplementary Information (Page S12, Table S3).

entry	HAT reagent	Results of 5a
1	(MeO) ₃ SiH (BDE(calc.) = 96.3 kcal/mol)	nd
2	Ph ₂ SiH ₂ (BDE = 90.6 kcal/mol)	nd
3	Ph ₃ SiH (BDE = 88.7 kcal/mol)	nd
4	(TMS) ₃ SiH (BDE = 79 kcal/mol)	56% yield, 92% e.e.

Reply to comments by Reviewer 2

We appreciate **Reviewer 2** for favorable comments!

- (1) Guo and co-workers report an enantioselective functionalization of unactivated alkene with benzoxazolyl acetate via electrochemically generated carbon-centered radicals. These radicals are stabilized by a chiral Ni catalyst, and subsequent enantioselective radical addition to alkenes—followed by hydrogen atom transfer (HAT), further oxidation, or reduction with hydrosilane—yields products of dehydrogenative allylation, alkenylation, and hydroalkylation, respectively. This methodology enables the formation of highly enantioriched products, including those bearing all-carbon quaternary stereocenters. The authors also provide detailed mechanistic insight into the chemo- and enantioselectivity based on cyclic voltammetry and DFT calculations.

Answer: We appreciate Reviewer 2 for the favorable comments and helpful suggestions! These comments are greatly valuable and helpful for revising and improving our manuscript. We have made all the necessary amendments as suggested in our revised manuscript and Supplementary Information.

- (2) However, similar transformation by the same group have already been reported (Refs. 34 & 35), and radical addition to alkene moieties are well-established in prior literature (Refs. 10–12). Therefore, the key novelty of this work appears to lie in the switchable radical termination strategy. This reviewer recommends that the authors clarify the individual novelty and significance of each transformation pathway—namely, dehydrogenative allylation, alkenylation, and hydroalkylation. A more explicit discussion will help readers better appreciate the scope and innovation of the work. This report may be perceived as merely a combination of a known radical reaction in the first half and a known termination reaction in the latter half. The reviewer believes it is essential to provide a clear and thorough discussion on what advantages this methodology offers.

Answer: We thank the Reviewer 2 for their thoughtful feedback on the novelty and significance of our work. The asymmetric radical addition to unactivated alkenes represents a long-standing challenge in synthetic chemistry. While pioneering studies (Refs. 10–12) have demonstrated radical additions to such alkenes, stereocontrol has remained elusive. Moreover, these electrochemical methods are predominantly confined to oxidative allylation. To date, achieving stereocontrolled electrochemical functionalization of unactivated alkenes—particularly for oxidative alkenylation and hydroalkylation—has been an unsolved problem. Crucially, the selective and stereodivergent synthesis of dehydrogenative allylation, alkenylation, and hydroalkylation products from identical starting materials has not been achieved. Building upon our previous work in dehydrogenative amination and denitrative alkylation (Refs. 34, 35), we

have now developed a versatile electrochemical system for chiral radical control. Distinct from our earlier studies, this work directly tackles the functionalization of unactivated alkenes. We herein report the first asymmetric addition of chiral catalyst bound radicals to unactivated alkenes, which systematically enables enantioselective dehydrogenative allylation, dehydrogenative alkenylation, and hydroalkylation from identical starting materials. This demonstrates the remarkable versatility of our catalytic system in exerting stereochemical command over diverse reaction pathways.

As suggested by Reviewer 2, we have incorporated the following discussion in the revised manuscript: “However, the stereocontrolled functionalization of unactivated alkenes remains a formidable challenge. Although electrochemical oxidative allylation has been demonstrated—as exemplified by the work of the Xu group^{10,12}(Refs. 10,12)—achieving analogous dehydrogenative alkenylation and hydroalkylation under electrolytic conditions continues to pose significant hurdles. More broadly, the development of electrochemical radical addition platforms capable of divergently accessing dehydrogenative allylation, alkenylation, and hydroalkylation from identical precursors with high chemoselectivity and stereocontrol represents a major unmet goal in synthetic chemistry. In this work, we report a general and enantioselective electrocatalytic system for the divergent functionalization of unactivated alkenes. This strategy enables dehydrogenative allylation, alkenylation, and hydroalkylation from a chiral nickel-bound radical intermediate with high stereoselectivity, broad functional group tolerance, and compatibility with diverse carbon nucleophiles (Fig. 1c). The asymmetric catalytic system provides a unified platform to systematically investigate competitive radical termination pathways under mild electrochemical conditions.”

(3) Furthermore, the origin of the selectivity, particularly between the allylation and alkenylation pathways, requires further clarification. According to the authors, the use of a Co catalysts leads to allylation, possibly due to steric hindrance influencing HAT. In contrast, oxidation followed by deprotonation results in alkylation. However, this reviewer finds the explanation for the selective deprotonation step insufficient. What governs the divergence between these two pathways? A more detailed mechanistic rationale, possibly supported by additional data or calculations, would greatly enhance the manuscript.

Answer: We thank the reviewer for this valuable suggestion. In response, we have performed additional computational studies to gain deeper mechanistic insights. These studies now provide a clearer rationale that explains the precise origin of the observed chemoselectivity between the competing allylation and alkenylation pathways.

Our mechanistic investigation establishes that the mediator **A1**-promoted alkenylation proceeds via an oxidation–radical addition–elimination sequence. Specifically, following the

initial radical addition, the resultant alkyl radical undergoes oxidation to a carbocationic intermediate. This key intermediate is stabilized by 2,6-lutidine (*Nat. Catal.* **2025**, 8, 448), and a subsequent base-promoted elimination furnishes the observed alkenylation product.

To elucidate the origin of chemoselectivity, we computed the key transition states for the competing alkenylation and allylation pathways. A comparative DFT analysis reveals that the alkenylation pathway (via **TS5-4a^s**) is energetically favored. The selectivity arises from the heightened acidity of the β -hydrogen atom, which is activated by three adjacent electron-withdrawing groups—the benzoxazole, ester, and fluorine substituents. This synergistic activation collectively enhances the acidity of the β -hydrogen, facilitates its elimination and directs the outcome toward alkenylation product **4a**.

As described in our response to Reviewer 2, the cobalt catalyst system selectively produces the allylation product **4a**. This selectivity is governed by steric effects in the hydrogen atom transfer (HAT) step, in which the bulky cobalt catalyst preferentially abstracts the less hindered distal hydrogen atom, thereby leading to the observed allylation outcome (Fig. 7c).

These computational insights now provide a quantitative basis for the divergence between the two pathways: the alkenylation route is governed by the relative acidity of specific C–H bonds and their propensity for elimination, while the allylation pathway is controlled by sterically modulated hydrogen atom transfer. We have incorporated these findings and the accompanying discussion into the revised manuscript “Conversely, the use of additive **A1** in the absence of additional termination reagents promotes single-electron oxidation of alkyl radical intermediate **III** to form carbocationic species **V**. This intermediate is stabilized by 2,6-lutidine and undergoes base-promoted β -hydride elimination preferentially from the more

acidic proximal carbon, ultimately furnishing enantioenriched dehydrogenative alkenylation products **4**.” and Supplementary Information (Page S61) to offer a more thorough and convincing mechanistic explanation.

(4) In all electrochemical reactions, the total passed (in F/mol) should be explicitly stated. Since external oxidants are replaced by anodic electron transfer, the charge of corresponding directly to the equivalent of oxidant used. This value can be calculated from the current and reaction time and should be reported for each transformation.

Answer: We have now calculated and explicitly stated the total charge passed (in F/mol) for each transformation in the revised manuscript. The total passed was determined to be 4.5 F/mol for both the allylation and alkenylation reactions, a value that aligns well with the electron equivalence provided by 2 equivalents of Ag₂O used in our control experiments. For the hydroalkylation reaction, the total passed was measured to be 6.0 F/mol. These values have now been clearly indicated in the corresponding sections of the main text.

(5) Page 1, line 25: “Main text” appeared to be a typographical error and should be corrected to “Introduction”

Answer: The term "Main text" on Page 1, line 25 has been corrected to "Introduction" in the revised manuscript.

(6) Page 8 line 199, The phrase “alcohol **7** with all-carbon quaternary stereocenters” is incorrect, as compound **7** does not possess an all-carbon quaternary stereocenter.

Answer: We have removed the phrase "with all-carbon quaternary stereocenters" from Page 8, line 199 in the revised manuscript.

(7) In the CV analysis, the authors note a significant decrease in oxidation potential upon addition of the Ni catalyst. However, the oxidation of morpholine also occurs at a similarly low potential (Figure 6a, top-left). To clearly distinguish the contribution of the Ni catalyst from that of morpholine, the cyclic voltammogram of a mixture of **1b** and Ni catalyst should be measured and presented.

Answer: To definitively assign their contributions, we acquired additional CV data for a mixture of **1b** and the nickel catalyst. Analysis of the complete CV dataset reveals a critical insight: a substantial decrease in the onset oxidation potential is observed specifically in the ternary system containing **1b**, morpholine, and the nickel catalyst. In contrast, binary combinations show no such effect. This evidence allows us to conclude that morpholine functions mainly as a Brønsted base to generate the enolate, while the nickel catalyst is central

to forming the catalytically active species that is readily oxidized.

Furthermore, we employed computational methods to evaluate the oxidation potentials of morpholine and the key nickel-bound enolate intermediate separately. The calculated oxidation potential for morpholine is approximately +1.26 V (vs. SCE), whereas that of the catalytic intermediate is about +0.46 V (vs. SCE). This significant difference clearly excludes any substantial interference from morpholine oxidation in the observed low-potential signal, confirming that the prominent anodic wave in the full system originates from the oxidation of the nickel-associated enolate species. We have included the cyclic voltammetry (Fig. 6a, left) in the revised manuscript.

Reply to comments by Reviewer 3

Answer: We thank the co-reviewer for their careful evaluation of our manuscript and their valuable contributions to the peer review process. We appreciate the time and expertise invested in assessing our work, and we have addressed all comments in our revised manuscript.

Reply to comments by Reviewer 4

We appreciate **Reviewer 4** for favorable comments!

- (1) The manuscript entitled “Enantioselective Switchable Selectivity in Electrocatalyzed Functionalization of Unactivated Alkenes” by Guo et al. describes the development of the asymmetric electrochemical α -functionalization of benzoxazolyl acetate derivatives with alkenes, via the use of a chiral Ni(II) complex. This method permits the transient formation of a chiral carbon-centered radical intermediate, able to diastereoselectively couple with alkenes to generate alkyl radicals which, depending on the reaction conditions, lead to the formation of alkenyl-, allyl-, or alkyl-functionalized products. The reaction outcome depends of the nature of the catalyst or the reagent introduced in the reaction mixture. Employing the ferrocene derivative A1 as redox mediator alone preferentially leads to the formation of the alkenylated products, where the cobalt complex A5 permits the selective preparation of the corresponding allylated derivatives. Alternatively, when A1 is employed in the presence of $(\text{Me}_3\text{Si})_3\text{Si-H}$ as terminal HAT reagent, the corresponding alkylated products are formed. The manuscript is very well written and the supporting information is of excellent quality. The development of enantioselective electrosynthetic transformation is of high interest in organic chemistry and is still a difficult goal to achieve. In this area, the use of chiral Lewis acids or chiral organocatalysts has recently emerged as the most promising catalytic strategies. In line with previous work from Guo et al. concerning the enantioselective electrochemical amination, α - and γ -Nitroalkylation, difunctionalization of terminal alkynes, and functionalization of alkenes, the strategy which is employed is based on the single electron oxidation of catalytically generated chiral nickel(II)enolate species, and the subsequent capture of the electrophilic radical which is formed by an alkene. Somehow, it can be regarded as a natural extension of previous work with one another type of radical quencher.

Answer: We appreciate Reviewer 4 for the favorable comments and helpful suggestions! These comments are greatly valuable and helpful for revising and improving our manuscript. We have made all the necessary amendments as suggested in our revised manuscript and Supplementary Information.

- (2) In the introduction of the manuscript, the authors emphasize the term “switchable,” seemingly as a way to highlight the versatility of their methodology in delivering different products. However, the use of this term may be somewhat misleading. In the context of catalysis, “switchable catalysis” typically refers to processes in which the reaction outcome can be altered by applying an external stimulus (e.g., light, heat, pH, or changes

in electronic state). In the present case, the different transformations— α -alkenylation, α -allylation, and α -alkylation—require distinct reaction conditions rather than a single system being modulated by an external factor. Therefore, the use of the term “switchable” may not be entirely appropriate in this context.

Answer: We thank the reviewer for this helpful suggestion. We have revised the manuscript accordingly, replacing the term "switchable" with "tunable" throughout the text. We agree that "tunable" is a more precise and appropriate descriptor for the process, as it accurately reflects how the product selectivity is modulated by varying specific reaction parameters.

(3) The optimizations of the reaction’s conditions are rather succinct; it would be more informative to discuss the exact role of the DCE/TFE solvent mixture.

Answer: We thank the reviewer for this insightful comment. The DCE/TFE solvent mixture was optimized to fulfill both electrochemical and chemical requirements. Neat DCE lacks sufficient conductivity for efficient electrolysis, while TFE not only provides the necessary conductivity but also serves as a proton source for the cathodic reaction. However, using TFE as the sole solvent led to a significantly lower yield (44%) compared to that achieved with the mixed solvent system (72% yield, 92% e.e.), which we attribute to its inherent acidity suppressing the crucial deprotonation step for enolate formation. Therefore, the DCE/TFE mixture achieves an optimal balance, ensuring adequate conductivity while maintaining a proper chemical environment for the key deprotonation event. These optimization results have been included in the Supplementary Information (Page S13, Table S6).

entry	Solvent (3 mL)	Results of 4a
1	DCE/TFE=0.5/2.5	72% yield, 92% e.e.
2	DCE	n.r.
3	TFE	44% yield, 92% e.e.

(4) why sometimes 2,6-lutidine is used and, sometimes, morpholine is preferred.

Answer: Our screening revealed that both 2,6-lutidine and morpholine can promote the alkenylation and allylation reactions while maintaining excellent enantioselectivity. However, we observed significant differences in reaction yields: for the allylation pathway, morpholine provided substantially higher yields compared to 2,6-lutidine, while the opposite preference was found for the alkenylation reaction. We therefore optimized the base selection specifically

for each transformation pathway to maximize the reaction efficiency. We have included these results in our revised Supplementary Information (Page S14, Table S7 and S8).

(5) How can we rationalize the fact that A5 is better than A2-A4,

Answer: Our DFT calculations provide an electronic structure basis for the observed catalyst performance, which reveal that the Co–H species generated from catalyst **A5** possesses a higher bond dissociation energy (BDE) than those from **A2–A4**. This higher BDE indicates a stronger hydrogen affinity, which enhances the efficiency of the hydrogen atom transfer (HAT) step. This computational insight correlates well with our experimental data, rationalizing the superior performance of **A5** in promoting the allylation pathway. We have included the following sentence to the revised manuscript: "The differential chemoselectivity among Co catalysts **A2–A5** correlates with their Co–H bond dissociation energies (BDE), where the higher BDE of **A5** enhances HAT efficiency and favors allylation product formation (See Supplementary Information)." The complete computational data have been included in the Supplementary Information (Page S60, Table S11).

entry	Co reagent	BDE(Co–H)/kcal mol ⁻¹
1	A2	35.4
2	A3	35.2
3	A4	35.5
4	A5	36.3

(6) what is the reactivity difference of A5 vs A1 (see below)?

Answer: The distinct roles of **A1** and **A5** stem from their fundamentally different coordination

geometries and associated mechanistic functions. **A1**, being coordinatively saturated, serves exclusively as an electron-transfer mediator. In the alkenylation pathway, it facilitates the oxidation of the Ni-bound enolate to generate the key radical intermediate but does not participate in the subsequent β -hydride elimination, which is instead mediated by the basic additive 2,6-lutidine. In contrast, catalyst **A5** possesses an accessible coordination site, enabling its direct involvement in the hydrogen atom transfer step during allylation. The extended planar geometry of its ligand further imposes steric control over the HAT regioselectivity. These combined attributes allow **A5** to selectively promote the allylation pathway, as consistently supported by experimental and computational evidence.

(7) With regard to the mechanistic studies performed, several important questions remain unanswered. First, why were all cyclic voltammetry (CV) experiments conducted in the presence of morpholine rather than lutidine? Lutidine is less prone to oxidation than morpholine, and this difference could significantly impact the CV results. It is worth noting that while lutidine is employed in both the alkenylation and alkylation reactions, morpholine is used exclusively in the Co-catalyzed allylation process.

Answer: We appreciate the reviewer 4's observation regarding the base selection in our CV studies. It is important to clarify that both 2,6-lutidine and morpholine can promote the alkenylation and alkylation reactions, with the primary distinction being their influence on the reaction yield. Our initial use of morpholine in CV measurements was motivated by its stronger basicity relative to 2,6-lutidine, which produced a more distinct catalytic current and facilitated clearer interpretation of the electrochemical response.

In response to the reviewer 4's comment, we have conducted additional CV experiments

using 2,6-lutidine. These results confirm the appearance of a catalytic current at a comparable onset potential, albeit with lower intensity than in the morpholine system. This finding aligns with the capacity of both bases to support the alkenylation and alkylation pathways, with their main differential influence lying in reaction yield. We have included these results in our revised Supplementary Information (Page S47, Figure S6).

- (8) Additionally, the experimental details for the CV measurements are insufficient. Key parameters such as the concentrations of analytes (as a result, the stoichiometry of the various analytes in the cell cannot be determined, which is critical considering the significant current measured), the solvent system, and the scan rate are not reported. Furthermore, the rationale for selecting compound **1b** as the model substrate instead of **1a** is not explained.

Answer: The full experimental parameters for the cyclic voltammetry (CV) measurements—including analyte concentrations, solvent composition, and scan rate—are provided in Section 7 (Cyclic Voltammetry Studies) of the Supplementary Information (Pages S45–48). All CV experiments were performed in an electrolyte of $n\text{Bu}_4\text{NPF}_6$ (0.05 M) using a DCE/TFE (2:1, 3.0 mL) solvent system with a scan rate of 100 mV/s. Regarding the choice of substrate **1b** over **1a** for the CV studies, preliminary experiments showed that **1a** gave a less distinct catalytic current under the same conditions. In addition, computational analysis indicated that the oxidation potentials of the corresponding nickel-bound α -carbonyl radical intermediates are similar for both substrates: +0.46 V vs. SCE for **1b** and +0.61 V vs. SCE for **1a**. Based on the electrochemical response observed experimentally with **1b**, it was selected as the model substrate for detailed CV characterization.

(9) More critically, the conclusions drawn by the authors are questionable. The oxidation wave attributed to the nickel(II) enolate appears very close to that of morpholine (Figure 6a, top left). Given that the reaction conditions involve a tenfold excess of morpholine relative to the chiral nickel catalyst (which is likely not the case for the CV measurements), competitive oxidation of morpholine cannot be ruled out. The earlier onset of the oxidation wave upon increasing the concentration of nickel catalyst (Figure 6a, top right) is not sufficient evidence to confirm the oxidation of the nickel enolate. The measured currents are very weak, and once again, it appears that only one equivalent of morpholine was used.

Answer: We have conducted additional experiments and computational studies to unequivocally address the possibility of interference from morpholine oxidation. To exclude potential interference from morpholine, we performed complementary CV experiments using 2,6-lutidine as the base. These measurements also revealed a distinct catalytic wave at comparably low potentials, confirming that the observed oxidation feature originates from a newly formed catalytic intermediate rather than from the oxidation of the base. Furthermore, computational studies provide independent support for this assignment. The calculated oxidation potential for morpholine is +1.26 V (vs. SCE), while that of the key nickel-bound enolate intermediate is +0.46 V (vs. SCE). This substantial difference suggests limited contribution from morpholine oxidation to the catalytic wave observed at lower potentials. These combined experimental and computational results imply that the oxidation wave at +0.46 V corresponds to the nickel-bound enolate species and is not affected by competitive oxidation of the base.

(10) Even more problematic is the interpretation of the role of the Co(II)-salen catalyst. The

authors suggest that the disappearance of the reverse peak indicates oxidation of the nickel enolate by the Co(III) complex via a single-electron transfer (SET) process, thereby generating a carbon-centered radical. However, the CV data do not support this conclusion. Specifically, increasing the concentration of the nickel catalyst does not result in a catalytic wave (Figure 6a, bottom right). If Co(III)+ generated electrochemically were indeed oxidizing the enolate via SET, a catalytic current—similar to that observed with compound A1—should be present. Since it is not, this suggests that a two-electron oxidation is taking place, or that a chemical transformation other than a simple electron transfer is occurring between the nickel enolate and Co(III)+.

Answer: We sincerely thank the reviewer for this insight, which has led us to reevaluate and significantly improve our Co-involved mechanistic understanding. The reviewer is correct that our original proposal of a SET process between the Co(III) complex and the nickel enolate was not adequately supported. Following the reviewer 4's suggestion, we performed control CV experiments in the absence of both substrate **1b** and base. Interestingly, varying the concentration of the Ni/L5 catalyst resulted in gradual attenuation of the reduction peak while the oxidation peak remained largely unchanged. This observation indicates limitations in our initial mechanistic proposal and suggests that the Co-salen species **A5** may not function primarily as an electron-transfer mediator for Ni-bound enolate species.

Furthermore, control reactions conducted in the absence of both ferrocene mediators and cobalt catalysts still afforded alkenylation product **4a** in 26% yield with 92% ee (Table 1b, entry 3). This result demonstrates that the nickel enolate intermediate can undergo direct anodic oxidation to generate the key Ni-bound α -keto radical species, without requiring SET oxidation by Co(III). We have incorporated these revisions and corresponding discussions (Fig. 6a) throughout the revised manuscript and Supplementary Information.

(11) This finding contradicts the authors' proposed mechanism and raises serious doubts about the validity of the allylation pathway as described. Consequently, the DFT calculations based on a SET mechanism are also questionable. In addition, the proposed regeneration of the Co(II) catalyst is presented inconsistently. In Figure 7a, it is suggested via DFT calculations to occur via comproportionation between Co(I) and Co(III), whereas Figure 7d instead depicts anodic oxidation as the operative step. This discrepancy should be clarified.

Answer: We have revised our computational model to reflect direct anodic oxidation of the nickel enolate, rather than Co(III)-mediated oxidation (Fig. 7a). The oxidation potential was set to +0.32 V vs. SCE, corresponding to the experimentally measured onset potential for the nickel-bound enolate species. The revised energy profile now shows an overall barrier of 20.0 kcal/mol at this potential, which falls within the expected range for room-temperature reactions (<21 kcal/mol) and aligns well with our experimental observations.

Based on our experimental and computational investigations, we have suggested that the salen-Co species does not function as an electron-transfer mediator in this system. Consequently, the Co(I) species is oxidized to Co(II) via direct anodic oxidation rather than through oxidation by Co(III). The relevant computational sections and proposed catalytic cycles have been revised accordingly in the revised manuscript.

(12) In conclusion, the manuscript by Guo et al. presents interesting findings but lacks sufficient novelty to be considered highly innovative. The work appears to be a natural extension of previous studies, employing alkenes to trap a transient chiral, carbon-centered radical intermediate. While the authors claim that the reaction conditions allow access to three types of products—alkenylated, allylated, or alkylated—the description of the process as "switchable" is misleading. Rather than a single switchable process, the study describes three distinct reactions proceeding via closely related radical pathways, with alkenes serving as radical traps (reminiscent of their previous work: *Sci. Adv.* 2022, 8, eadd7134). The fate of the transient alkyl radical follows well-established transformations, including oxidation/elimination, oxidative hydrogen atom transfer (HAT) with a cobaloxime-type catalyst, and reductive HAT using $(\text{Me}_3\text{Si})_3\text{Si-H}$. However, the proposed mechanism is questionable, particularly regarding the role and mode of action of the Co(II)-salen catalyst. Cyclic voltammetry (CV) experiments suggest that the proposed single-electron transfer (SET) oxidation of the nickel enolate by electro-generated Co(III)⁺ is unlikely.

Answer: While radical additions to alkenes are indeed known, exerting excellent stereocontrol over these processes—particularly for unactivated alkenes—remains a formidable challenge in

synthetic chemistry. The key novelty of our work lies not merely in the use of an alkene as a radical trap, but in the development of a unified chiral catalytic system that enables, for the first time, enantioselective radical additions to unactivated alkenes with tunable termination pathways. The stereochemical control of open-shell intermediates at unactivated alkenes is inherently difficult due to the short-lived nature of radical species and the lack of pre-coordination sites. Our system addresses this by leveraging a chiral nickel catalyst to generate and manage the transient radical within a defined chiral environment, achieving high enantioselectivity across multiple pathways—a capability not previously demonstrated.

We acknowledge the reviewer 4's valid concerns regarding the initial mechanistic proposal for the Co-salen catalyst. We have thoroughly revised this aspect, removing the SET oxidation pathway as the mediator and demonstrating through control experiments and revised computations that the nickel enolate is instead directly oxidized at the anode, while the cobalt catalyst primarily functions in the HAT step for the allylation pathway.

In summary, the advance presented here is the integration of electrochemical generation, enantioselective radical trapping, and tunable termination into a single platform for unactivated alkene functionalization—addressing a long-standing challenge in stereoselective synthesis.

Reply to comments by Reviewer 5

Answer: We thank the co-reviewer for their careful evaluation of our manuscript and their valuable contributions to the peer review process. We appreciate the time and expertise invested in assessing our work, and we have addressed all comments in our revised manuscript.

Reply to comments by Reviewer 6

(1) In this manuscript, Guo et al. reported an enantioselective switchable electrolytic system for the functionalization of unactivated alkenes, which enables dehydrogenative allylation, dehydrogenative alkenylation, and hydroalkylation reactions with identical substrates to produce structurally diverse products. The success of this system relies on the stereocontrolled coupling of an electrogenerated Ni-bound α -carbonyl radical with alkenes, followed by divergent radical termination pathways. The method operates under mild, oxidant-free conditions, exhibits broad substrate scope (63 examples), and delivers high enantioselectivity (up to 98% ee). Applications include late-stage functionalization of complex molecules and a formal synthesis of (S)-SYK inhibitor, highlighting its synthetic

utility. Mechanistic studies and DFT calculations support the proposed catalytic cycle. The following issues should be considered:

Answer: We appreciate Reviewer 6 for the favorable comments and helpful suggestions! These comments are greatly valuable and helpful for revising and improving our manuscript. We have made all the necessary amendments as suggested in our revised manuscript and Supplementary Information.

(2) In the optimization of switchable reaction conditions, the authors mentioned the chemoselectivity of the reaction is strongly dependent on additive selection. It could be seen that the additive A1 with any ligand could form the alkenylated product 4a, while the presence of salen-Co additives could achieve the allylated product 3a. This is really interesting. DFT calculations partly explain the role of the Co additives, the additive A1 should also be calculated to clarify the chemoselectivity of this reaction.

Answer: Our investigation reveals that the alkenylation pathway, mediated by the **A1** mediator, proceeds through an oxidation–radical addition–elimination mechanism. Following radical addition, the resulting alkyl radical is oxidized to a carbocation intermediate, which is stabilized by 2,6-lutidine (*Nat. Catal.* **2025**, 8, 448). Subsequent base-promoted elimination then produces the alkenylation product.

To elucidate the chemoselectivity, we computed the key transition states leading to both alkenylation and allylation products. DFT results clearly indicate a pronounced energetic preference for the alkenylation pathway (**TS5-4a^s**). This selectivity can be attributed to the enhanced acidity of the hydrogen atom situated near the three electron-withdrawing groups—the benzoxazole, ester, and fluorine substituents—which facilitates its elimination and directs the reaction toward alkenylated product **4a**.

The use of a Co catalysts leads to allylation, due to steric hindrance influencing HAT, where the cobalt catalyst selectively abstracts the less hindered distal hydrogen atom, leading to the observed allylation product **4a** (Fig. 7c). These computational insights now provide a quantitative basis for the divergence between the two pathways: the alkenylation route is governed by the relative acidity of specific C–H bonds and their propensity for elimination, while the allylation pathway is controlled by sterically modulated hydrogen atom transfer. We have incorporated these findings and the accompanying discussion into the revised manuscript “Conversely, the use of additive **A1** in the absence of additional termination reagents promotes single-electron oxidation of alkyl radical intermediate **III** to form carbocationic species **V**. This intermediate is stabilized by 2,6-lutidine and undergoes base-promoted β -hydride elimination preferentially from the more acidic proximal carbon, ultimately furnishing enantioenriched dehydrogenative alkenylation products **4**.” and Supplementary Information (Page S61) to offer a more thorough and convincing mechanistic explanation.

(3) It would be better if the different effects of salen-Co additives for the diversity in ratio of different chemoselective products could be explained.

Answer: We propose that the observed product distribution arises from competition between the cobalt-catalyzed allylation pathway and the background alkenylation reaction (Table 1b, Entry 3). The variation in catalytic efficiency among different salen-Co complexes (**A2–A5**) appears to be closely related to their hydrogen atom transfer (HAT) capabilities. DFT calculations of the Co–H bond dissociation energies (BDE) in the corresponding salen-Co–H species reveal that the Co–H BDE for complex **A5** is 36.3 kcal/mol, higher than those calculated for **A2–A4**. This higher BDE indicates stronger hydrogen affinity of the **A5** catalyst, which facilitates more efficient HAT. Consequently, **A5** exhibits superior performance in promoting the formation of allylation product **3a**, thereby leading to enhanced chemoselectivity toward the allylation pathway. We have included these results in our revised Supplementary Information (Page S60, Table S11).

entry	Co reagent	BDE(Co–H)/kcal mol ⁻¹
1	A2	35.4
2	A3	35.2
3	A4	35.5
4	A5	36.3

(4) The ratios of E/Z product was mentioned in this manuscript, it would be great if the authors could also consider the possibility of E/Z product formation in the reaction.

Answer: To elucidate the origin of the stereochemical outcome, we computed the key transition states governing the formation of *E*- and *Z*-alkenes in both the allylation and alkenylation pathways. For the allylation reaction, the energy difference between transition states **TS3-3a^{s(BS)}** (*E*-pathway) and **TS3-3a-*Z*^{s(BS)}** (*Z*-pathway) is relatively small ($\Delta\Delta G^\ddagger = 0.4$ kcal/mol), corresponding to an E/Z ratio of approximately 6:1, which aligns well with our experimental observations. In contrast, for the alkenylation pathway, a significantly larger energy difference ($\Delta\Delta G^\ddagger = 4.6$ kcal/mol) is observed between **TS5-4a^s** (*E*-pathway) and **TS5-4a-*Z*^s** (*Z*-pathway). This substantial barrier rationalizes why the *Z*-isomer was not detected experimentally in the alkenylation products. Structurally, the bulky 2,6-lutidine base introduces considerable steric hindrance that disproportionately destabilizes the transition state leading to the *Z*-alkene, particularly in the alkenylation mechanism where the geometry around the forming double bond is more constrained. In comparison, the more planar and open coordination environment of the Co(salen) catalyst in the allylation pathway results in a lower energetic penalty for *Z*-isomer formation. We have included these results in our revised Supplementary Information (Page S62).

(5) Did the authors consider the spin states of Co and Ni in the DFT calculation? The authors only mentioned “For transition states TS-3a and TS-4a, the ‘broken symmetry’ formalism was used” in Supporting Information, have the authors tried and compared different spin states for the key stationary points? It is weird that different spin states were used for stationary points along reaction pathways in the calculation without additional explanation. This should be described in the manuscript including Figure 7(a).

Answer: We have considered the possible spin states of cobalt and nickel species present in the system. The spin-state changes occurring along the reaction pathway involve well-characterized species whose spin state are either documented in prior studies or can be reliably assigned via theoretical analysis (*J. Am. Chem. Soc.* **2024**, *146*, 34043; *Nat. Commun.* **2017**, *8*, 14875; *Inorg. Chem.* **2012**, *51*, 10557; *Angew. Chem. Int. Ed.* **2025**, e202506268). Specifically, six-coordinate octahedral nickel (II) complexes adopt a high-spin configuration; cobalt centers coordinated by planar quadridentate ligands (such as Salen) favor a low-spin state; and the carbon radical generated during the reaction exhibits weak coupling with nickel, making the spin-parallel configuration slightly more stable than the spin-antiparallel (broken symmetry) configuration.

To further validate the chosen spin states, we computed the Gibbs free energies difference between the spin state we have chosen and another plausible spin state for the cobalt and nickel complexes. Given that accurately predicting spin-state energy differences remains challenging within DFT, we performed single-point calculations using the TPSSh-D3(BJ) and MN15L functionals—selected based on benchmark studies in the literature (*J. Chem. Theory Comput.* **2020**, *16*, 4416; *Inorg. Chem.* **2018**, *57*, 14097)—instead of the wB97X-D functional. In all cases, the energy differences between the spin states employed in our study and other plausible spin states were positive, confirming our choice correspond to the most stable spin states.

entry	Intermediate	The chosen spin state to another plausible spin state	$\Delta G(\text{TPSSh-D3(BJ) Single-Point})$	$\Delta G(\text{MN15L Single-Point})$
1	$[\text{Ni}^{\text{II}}]^+\text{OAc}^-$	t \rightarrow s	15.3	21.8
2	INT1	t \rightarrow s	14.1	20.4
3	INT2	t \rightarrow s	15.3	22.2
4	INT3	q \rightarrow d(BS)	1.0	2.1
5	INT4	q \rightarrow d(BS)	0.4	0.4
6	$[\text{Co}^{\text{II}}]$	d \rightarrow q	6.9	0.5
7	INT6	s \rightarrow t	17.8	15.6
8	$[\text{Co}^{\text{I}}]$	s \rightarrow t	7.0	2.1

The spin states of the transition states were logically deduced from those of the adjacent intermediates. For TS3, which corresponds to the hydrogen atom transfer (HAT) step, the reactants are two radical species, while the product is a closed-shell system. For such electron pairing processes, employing the broken-symmetry (BS) approach in DFT calculations is both appropriate and necessary (*Chem. Phys. Lett.* **2000**, 319, 223).

Additionally, following the reviewers' suggestions, we have added superscripts representing the spin states in Figure 7(a): "Spin states are annotated as superscripts: s (singlet), d (doublet), t (triplet), q (quartet), and BS (broken symmetry)."

(6) The origin of stereoselectivity could be explained by NCI or EDA analysis. It is hard to understand TS2-Si is more favorable from the structures. Considering that TS2 is the key transition state structure that determines stereoselectivity, the discussion on its conformation in Figure S8 should be briefly mentioned in the manuscript.

Answer: We performed energy decomposition analysis on the energy difference between the key enantioselectivity-determining transition states **TS2-Si^q** and **TS2-Re^q**. Initial distortion–interaction analysis revealed that the transition state energy difference originates primarily from the distortion energy of the nickel-bound radical specie (Fragment 1, 2.9 kcal/mol), whereas both the distortion energy of **2a** (hexene, Fragment 2, -0.9 kcal/mol) and interaction energy (-0.6 kcal/mol) disfavor enantioselectivity. This suggests that the origin of enantioselectivity is likely governed by the conformation of the nickel-bound radical specie.

Further distortion–interaction analysis focusing on the nickel-bound radical specie (Fragment 1) indicated that the energy difference arises mainly from the distortion energy of the nickel catalyst (Fragment 3, 1.4 kcal/mol), followed by contributions from the distortion energy of benzoxazolyl acetate radical (Fragment 4, 0.7 kcal/mol) and interaction energy (0.9 kcal/mol). The result implies that the energy difference of the nickel-bound radical specie stem from distinct benzoxazolyl acetate substrate coordination modes, leading to varied steric repulsion and consequently significant nickel catalyst distortion coupled with benzoxazolyl acetate substrate distortion.

distortion-Interaction analysis of TS2

distortion-Interaction analysis of Fragment1

energy decomposition analysis of E_{int} (Fragment1)

Since interaction energy E_{int} (0.9 kcal/mol) contributes non-negligibly to the energy difference, we conducted additional energy decomposition analysis, which identified electrostatic interactions as the dominant component (5.6 kcal/mol), with orbital interactions playing a minor role (1.6 kcal/mol). Consistent with our earlier findings (*J. Am. Chem. Soc.* **2024**, *146*, 34043), the two nickel-bound radical specie exhibit a noticeable difference in N–Ni bond lengths (slightly longer in the **TS2- Re^q** complex, **TS2- Si^q** : 2.08 Å; **TS2- Re^q** : 2.10 Å), whereas the O–Ni distances remain similar (**TS2- Si^q** : 2.13 Å; **TS2- Re^q** : 2.13 Å). The coordination difference correlates well with the larger electrostatic energy difference and smaller orbital interaction energy difference. We thus attribute the interaction energy difference of the nickel-bound radical specie to the weaker N–Ni coordination in **TS2- Re^q** .

Structural analysis suggests that, under the **TS2- Re^q** coordination mode, steric repulsion occurs between the phenyl ring of the benzoxazolyl acetate substrate and the BOX ligand. This structural distinction consistently accounts for the greater distortion energy difference of the nickel catalyst and the smaller distortion energy difference of the benzoxazolyl acetate substrate in **TS2- Re^q** . Furthermore, it leads to impaired contact between the coordinating N atom (adjacent to the phenyl substituent) and the Ni center, thereby resulting in a weaker N–Ni bond, whereas the O–Ni bond remains unaffected.

Integrating the results from distortion-interaction analysis, energy decomposition analysis, and structural analysis, we conclude that the enantioselectivity arises from energy differences

induced by distinct substrate coordination modes. In particular, the coordination in **TS2-*Re*^a** introduces steric repulsion between the phenyl substituent of the benzoxazolyl acetate substrate and the BOX ligand, resulting in a disfavored transition state.

In response to the reviewer's suggestion, we have incorporated a discussion on the origin of enantioselectivity in the revised manuscript, including a brief description of Figure S8 (Figure S9 in revised Supplementary Information). Our analysis reveals that the enantioselectivity primarily stems from distinct coordination modes of the benzoxazolyl ester substrate. These different coordination modes lead to opposing facial selectivities, as illustrated in Figure S8. Further energy decomposition analysis (see the Supplementary Information) indicates that the variation in coordination geometry arises mainly from steric repulsion between the phenyl ring of the substrate and the BOX ligand.

(7) The structure of **2a** should be given in Figure 7.

Answer: The structure of compound **2a** has been added to Figure 7 in the revised manuscript.

Responses to Comments

for

Tunable Enantioselective Electrocatalytic Functionalization of Unactivated Alkenes

Tian Xie, Minghao Liu, Jiayin Zhang, Lingzi Peng, Chang Guo 
Manuscript ID: NCOMMS-25-46490A

Reply to comments by Reviewer 1

(1) All my concerns have been addressed and I does not have further concerns.

Answer: We appreciate **Reviewer 1** for favorable comments!

Reply to comments by Reviewer 2

We appreciate **Reviewer 2** for favorable comments!

(1) I have reviewed the revised version of the manuscript. Overall, the authors have addressed the points raised in the previous review, and the manuscript is much improved. Although the cyclic voltammetry experiments may require further careful consideration, as also noted by Reviewer 4, this reviewer looks forward to further studies by the authors on the radical generation mechanism. Therefore, this reviewer is happy to recommend the manuscript for publication as is.

Answer: We appreciate Reviewer 2 for the favorable comments and helpful suggestions! Following this advice, we have performed additional CV experiments and included the revised data as Figure 6 in the revised manuscript.

Reply to comments by Reviewer 3

Answer: We thank the co-reviewer for their careful evaluation of our manuscript and their valuable contributions to the peer review process.

Reply to comments by Reviewer 4

We appreciate **Reviewer 4** for favorable comments!

- (1) After proofreading this manuscript for the second time, I acknowledge that this new version has been significantly improved and has addressed most of my previous comments. Notably, the revised mechanistic proposal, excluding any role of Co(III) as a redox mediator in the oxidation of the Ni-enolate through a SET process, is more reasonable. The analyses and DFT calculations now better support the observed selectivity.

Answer: We appreciate Reviewer 4 for the favorable comments and helpful suggestions! These comments are greatly valuable and helpful for revising and improving our manuscript. We have made all the necessary amendments as suggested in our revised manuscript and Supplementary Information.

- (2) However, in my opinion, several aspects of the electrochemical studies remain unclear. To ensure relevant electrochemical measurements, the supporting electrolyte should be present in large excess compared with the substrates. In the current work, the concentration of the supporting electrolyte (50 mM) is only about twice that of 1b and the amine (33 mM), which is insufficient. Increasing the supporting electrolyte concentration could improve, even change, the CV profiles. The concentrations used for each component are too high. Considering the relatively large electrode (3 mm diameter), it would be more appropriate to present current values in microamperes (μA) rather than amperes (A). In the present case, the reported currents appear low but should be around 10 μA , which is much more realistic than the 0.3 mA values shown.

Answer: As suggested by Reviewer 2, we have increased the concentration of the supporting electrolyte from 50 mM to 100 mM and reduced the concentration of the substrates from 33

mM to 10 mM. As a result, the electrolyte is now in a 10-fold excess relative to the substrates. Under these conditions, we observed only minor changes in the CV measurements. Additionally, the current values have been converted to microamperes (μA) for better clarity. The corresponding CV profile (Fig. 6a) and the related figures in the Supplementary information (Figures S1–S7) have been updated accordingly.

- (3) Regarding the determination of onset potentials in the Supporting Information, this point deserves closer examination, as it is used to validate the authors' mechanistic discussion. The onset potential for morpholine appears to be below 1.16 V. The onset should be determined at the point where the current deviates from zero — with such high currents, demonstrating oxidation should not be problematic.

Answer: We thank reviewer 4 for this comment. In accordance with the suggestion, the onset potentials have been re-determined following the method described in *ACS Catal.* **2023**, *13*, 2916, where the onset is identified as the potential at which the current deviates from the baseline. All onset potentials have been revised accordingly, and corresponding updates have been made in the main text (including the DFT calculations in Fig. 7a) as well as in the Supplementary Information (Figures S1–S5).

- (4) The cyclic voltammograms (CVs) presented are not entirely convincing due to the excessively high currents observed for the 1b + morpholine mixture. A more convincing approach would be to record CVs with increasing amounts of this mixture added to a 1–2 mM Ni/L5 solution, yielding current intensities in the μA range.

Answer: Accordingly, we have performed CV measurements by adding increasing amounts (2–10 mM) of the **1b** and morpholine mixture to a 2 mM Ni/L5 solution. The results clearly show a significant increase in the catalytic current upon addition of the substrate and base, supporting our previous conclusion that the catalytic current arises from Ni-bound substrate oxidation and occurs at relatively low potentials. These new CV profiles have been included as the middle panel in Fig. 6a of the revised manuscript.

- (5) The authors state that “the initial oxidative potential of benzoxazolyl propanoate **1b** decreased significantly from +1.92 V to +0.32 V (versus SCE) upon addition of the chiral nickel catalyst.” Such comparisons are inappropriate; instead, the oxidation potential of the enolate (**1b** + morpholine) should be compared to that of the Ni-enolate. In this respect, the black arrow in Figure 6a is confusing.

Answer: We agree with the reviewer 4 that a direct comparison of potentials between the free substrate and the Ni-enolate is not appropriate. To address this, we have removed the black arrow. The revised text now focuses on the key observation—the negative shift of the oxidative peak upon catalyst addition. We have revised “Upon addition of the chiral nickel catalyst, the oxidative peak of benzoxazolyl propanoate **1b** shifted negatively (Fig. 6a, left) and exhibited a concurrent increase in current with higher concentrations of **1b** (Fig. 6a, middle), supporting the facile oxidation of a nickel-bound enolate intermediate.” in our revised manuscript.

- (6) Moreover, the CVs obtained with increasing catalyst concentrations are not relevant and, in my opinion, should be removed from the main text. Under the reaction conditions, only about 10 mol% of the nickel enolate is present alongside ten equivalents of the enolate; thus, the middle panel of Figure 6a does not provide meaningful information and should likely be omitted.

Answer: In accordance with the suggestion, we have removed the CV profiles obtained with increasing catalyst concentrations from Fig. 6a (right). Furthermore, the middle panel of Fig. 6a has been replaced with new CV measurements performed by adding increasing amounts (2–10 mM) of the substrate **1b** and morpholine to a 2 mM Ni/L5 solution, which directly reflects the catalytic reaction conditions and shows the role of the nickel catalyst in facilitating the oxidation.

- (7) The combination of only two reaction partners shown in the rebuttal is very informative. However, the CV of 1b with Ni alone was not presented. It would be helpful to include this binary combination, in the Supporting Information.

Answer: We thank the reviewer 4 for the constructive suggestion. Accordingly, we have included the CV profile of the binary combination (**1b** + Ni/**L5**) in the updated Fig. 6a (left) to provide additional mechanistic insight. The relevant figure in the manuscript has been revised accordingly.

- (8) The authors claim that both 2,6-lutidine and morpholine can promote the alkenylation and alkylation reactions, yet CV data were only provided for morpholine. They justify this by stating that morpholine affords a more distinct catalytic current and clearer interpretation of the electrochemical response. However, no catalytic current is observed with either morpholine or 2,6-lutidine in the comparison CVs shown in the rebuttal. This explanation is therefore unconvincing and should be reconsidered.

Answer: We appreciate the reviewer 4's comment. It is indeed challenging to observe a strong oxidative peak for the Ni-enolate intermediate due to its high reactivity and low concentration. This is supported by DFT calculations, which indicate that the formation of the Ni-enolate intermediate (**INT2**) is energetically uphill ($\Delta G = +1.8$ kcal/mol), underscoring its inherent instability. We have further investigated the influence of different bases on the CV behavior of

the system. The onset oxidation potential of the Ni-enolate intermediate remains nearly identical regardless of whether morpholine or 2,6-lutidine is used as the base. Upon magnification of the CV curves, a clear catalytic current can be observed, corresponding to the oxidation of the Ni-enolate species. Moreover, in control experiments performed in the absence of the nickel catalyst, no such catalytic current is observed with either base. All relevant CV profiles have been included in the Supporting Information (Figures S4–S5), and these consistent findings support the proposed mechanistic pathway involving the nickel-enolate species.

(9) The only compelling observation is that displayed in Figure 6a (right), which shows that **A1** can act as a redox mediator. Nevertheless, this role is not integrated into the mechanism proposed in Figure 7.

Answer: We thank the reviewer 4 for this valuable observation. Figure 6a (right) demonstrates that ferrocenemethylamine **A1** functions as an effective redox mediator in the alkenylation reaction. Under the revised experimental conditions, we have reconfirmed that **A1** undergoes efficient electron transfer specifically with the nickel-enolate catalytic intermediate. It is

important to clarify that the mechanism outlined in Figure 7d describes the Ni/Co-catalyzed oxidative allylation system, which operates in the absence of a ferrocene mediator. Therefore, the role of **A1** as a redox mediator has been explicitly addressed in the context of Fig. 6a. The corresponding description in the main text has been updated as follows: “Critically, while the substrate alone with ferrocenemethylamine **A1** showed no significant catalytic current, the addition of the chiral nickel catalyst led to a substantial increase in the oxidative current with attenuation of the reduction peak. This clear contrast provides evidence for efficient electron transfer between **A1** and the Ni-bound enolate intermediate (Fig. 6a, right).”

(10) It would be useful to perform CVs under conditions closer to the catalytic system — namely with a 1:1 mixture of **A1** and Ni/**L5**, along with ten equivalents of **1b** and 2,6-lutidine — rather than the conditions used in Figure S8, which are not representative.

Answer: We thank the reviewer for this constructive suggestion. Using a 1:1 mixture of **A1** and Ni/**L5** along with ten equivalents of **1b** and 2,6-lutidine—we have repeated the CV measurements (presented in the revised Figure S8). The results clearly demonstrate that introduction of the chiral nickel catalyst leads to a catalytic current, which confirms electron transfer between **A1** and the catalytic intermediate. By contrast, no such current is detected in the absence of the nickel catalyst, underscoring the essential role of the nickel catalyst in facilitating the oxidative process.

(11) In conclusion, since the stereocontrolled electrochemical functionalization of unactivated alkenes remains a largely unexplored and challenging area, I believe this work is of significant interest to the scientific community. However, the electrochemical data supporting the proposed mechanism require further improvement. For these reasons, I recommend publication in Nature Communications once the authors have thoroughly addressed all the issues mentioned above.

Answer: We sincerely thank Reviewer 4 for their encouraging conclusion and insightful suggestions, which have been invaluable in strengthening the electrochemical support for our proposed mechanism. All raised issues have been fully addressed in the revised manuscript and supplementary information, leading to a significant improvement in the quality and clarity of our data and discussion.

Reply to comments by Reviewer 5

Answer: We thank the co-reviewer for their careful evaluation of our manuscript and their valuable contributions to the peer review process.

Reply to comments by Reviewer 6

(1) The authors have revised the manuscript according to my comments, it could be acceptable now.

Answer: We appreciate Reviewer 6 for the favorable comments and helpful suggestions!

Responses to Comments

for

Tunable Enantioselective Electrocatalytic Functionalization of Unactivated Alkenes

Tian Xie, Minghao Liu, Jiayin Zhang, Lingzi Peng, Chang Guo [✉]

Manuscript ID: NCOMMS-25-46490B

Reply to comments by Reviewer 4

We appreciate **Reviewer 4** for favorable comments!

- (1) I have examined the revised version of the manuscript, which now includes additional cyclic voltammetry (CV) experiments. The authors have satisfactorily addressed most of the concerns raised in the previous review, and the overall quality of the manuscript has improved considerably. Nonetheless, in their rebuttal letter, the authors continue to use the term “catalytic current,” which is not appropriate in the present context. Demonstration of a genuine catalytic process would require a much more pronounced effect, rather than the modest increase in oxidation peak intensity accompanied by an almost negligible decrease in the reduction peak. In any case, since the revised manuscript no longer refers to a catalytic current, this issue can be considered resolved.

Answer: We thank the reviewer 4 for thoughtful feedback and for acknowledging the improvements made to the manuscript. We agree that the term “catalytic current” was imprecise in this context, and we have changed “catalytic current” into “current response” in the revised manuscript. We appreciate the reviewer 4’s attention to this detail.

Reply to comments by Reviewer 5

- (1) I co-reviewed this manuscript with one of the reviewers who provided the listed reports. This is part of the Nature Communications initiative to facilitate training in peer review and to provide appropriate recognition for Early Career Researchers who co-review manuscripts.

Answer: We thank the co-reviewer for their careful evaluation of our manuscript and their valuable contributions to the peer review process.